# Single-cell RNA-sequencing analysis of the developing mouse inner ear identifies molecular logic of auditory neuron diversification

Charles Petitpré[1,6], Louis Faure [2,6], Phoebe Uhl [1], Paula Fontanet [1], Iva Filova[3], Gabriela Pavlinkova[3], Igor Adameyko [2,4], Saida Hadjab [1,7✉] & Francois Lallemend [1,5,7✉]

Different types of spiral ganglion neurons (SGNs) are essential for auditory perception by transmitting complex auditory information from hair cells (HCs) to the brain. Here, we use deep, single cell transcriptomics to study the molecular mechanisms that govern their identity and organization in mice. We identify a core set of temporally patterned genes and gene regulatory networks that may contribute to the diversification of SGNs through sequential binary decisions and demonstrate a role for NEUROD1 in driving specification of a $I_c$-SGN phenotype. We also find that each trajectory of the decision tree is defined by initial co-expression of alternative subtype molecular controls followed by gradual shifts toward cell fate resolution. Finally, analysis of both developing SGN and HC types reveals cell-cell signaling potentially playing a role in the differentiation of SGNs. Our results indicate that SGN identities are drafted prior to birth and reveal molecular principles that shape their differentiation and will facilitate studies of their development, physiology, and dysfunction.

[1] Department of Neuroscience, Karolinska Institutet, Stockholm, Sweden. [2] Department of Neuroimmunology, Center for Brain Research, Medical University Vienna, 1090 Vienna, Austria. [3] Institute of Biotechnology CAS, 25250 Vestec, Czech Republic. [4] Department of Physiology and Pharmacology, Karolinska Institutet, Stockholm, Sweden. [5] Ming-Wai Lau Centre for Reparative Medicine, Stockholm Node, Karolinska Institutet, Stockholm, Sweden. [6]These authors contributed equally: Charles Petitpré, Louis Faure. [7]These authors jointly supervised this work: Saida Hadjab, Francois Lallemend. ✉email: saida.hadjab@ki.se; francois.lallemend@ki.se

The ability to detect and discriminate auditory stimuli depends on sensory coding of sound components realized in the cochlea by the spiral ganglion neurons (SGNs). These primary auditory neurons receive synaptic input from the auditory hair cells (HCs, the sensory receptors located in the organ of Corti) at their distal dendrites and convey all auditory information to the cochlear nuclei in the brainstem through their central projections that form the auditory nerve. Recent molecular classifications have identified at least four different types of SGNs in mice ($I_a$-, $I_b$- and $I_c$-SGNs and II-SGNs)[1–3], consistent with the physiological and anatomical diversity of primary auditory afferents[4,5]: I-SGN fibers exhibit large differences in their spontaneous discharge rates and sensitivities, which correlate with the location and structural features of their synaptic contact at the base of the IHC. Such diversity of I-SGN fibers makes them capable of collectively span all levels of sound intensity. Therefore, abnormal differentiation and connectivity of the four SGN types is expected to disrupt their functional organization and thereby critically impact auditory function. However, when and how SGN subtype identities are established and the transcriptional mechanisms by which they emerge are currently unknown.

SGNs are produced in a base to apex progression from neuroblasts that have delaminated from the otocyst[6] and migrated to form a dense ganglion along the medial side of the inner ear sensory epithelium between E10 and E12 in mice[7]. As the cochlear duct elongates, these early SGNs extend peripheral projections through the expanding otic mesenchyme (OM) population to reach the developing sensory epithelium around E15-17[8,9]. Thereafter, SGN projections continue to grow within the cochlear epithelium to form synaptic connections with the HCs[10]. Simultaneously, the central projections of SGNs reach the cochlear nuclei[11], making contact with neurons of the central auditory pathway. Slightly before their innervation by SGNs, HCs differentiate into inner HCs (IHCs) and outer HCs (OHCs)[12,13] and complex nerve endings below OHCs suggesting differentiation of type II SGNs (II-SGNs)-like dendrites begin to be observed during late embryogenesis[11]. At birth, the characteristic morphologies of type I SGN (I-SGN) and II-SGN afferents below the HCs are clearly apparent. Accordingly, multiple I-SGNs innervate radially one IHC—which in the adult transduces the physical energy of sound into electrochemical signals—while, by contrast, single II-SGNs form *en passant* synaptic connections with a dozen of OHCs that in the mature cochlea modulate the sensitivity and therefore indirectly the output of the IHCs. Moreover, already by this stage, SGNs exhibit a large diversity of molecular profiles, varying connectivity patterns and distinct intrinsic physiological properties which to some degree prefigure the functional organization evident in the adult[1,14]. An unresolved question is therefore to understand the timing and diversity of the molecular events that produce and assemble sensorineuronal specificity in the developing cochlea and eventually enable animals to detect and respond to a plethora of auditory stimuli.

In this study, we used single-cell RNA-sequencing (scRNAseq) to define the molecular logic by which SGN subtypes diversify in the mouse. Our detailed analysis of transcriptional dynamics of SGNs from several embryonic and perinatal stages indicates that neuronal subtypes successively emerge during HC afferent innervation and provides a comprehensive collection of molecular states and candidate gene regulatory networks associated with each lineage and underlying fate divergence. Importantly, we uncover the functional consequence of *Neurod1* deletion on the first binary decision between the presumptive $I_c$-SGNs and the rest of the neuronal lineage. Finally, we catalogue chemotropic signaling that would define the complex cochlear wiring and identify deafness genes associated with distinct cell states of developing SGNs and HCs.

## Results

**Embryonic emergence of SGN diversity.** To analyse the molecular changes associated with the diversification of SGN and HC types during development, we used flow cytometry to sort tdTomato$^+$ (TOM$^+$) cells isolated from E14.5, E15.5, E16.5, E17.5 and E18.5 (Fig. 1a) and from P3 (previously published)[1] cochlea of *Ntrk3$^{Cre}$;R26$^{tdTOM}$* or *PV$^{Cre}$;R26$^{tdTOM}$* mice and sequenced their mRNA with high coverage using the Smart-seq2 protocol[15]. The selected stages cover the earliest time point (P3) SGN subtypes have been defined transcriptionally[1] as well as a key period in the embryonic development of the neurosensory elements of the cochlea characterized by the innervation of HCs by SGNs[11,16].

A total of 2308 cells were pre-processed and clustered with the pagoda2 pipeline. We applied Harmony[17], an algorithm to bridge time points, and combined it with Palantir[18], to generate a diffusion space which was then used as a basis for Force Atlas 2 (FA) embedding and for subsequent trajectory analysis (see "Methods"). The dynamic of the gene expression on our multidimensional integrated dataset identified 19 clusters (Cl., Fig. 1a), including intermediate states with varying degrees of cell fate biases/lineage restrictions of the SGNs, HCs and OM compartments (Fig. 1b). The neuronal compartment was characterized by the expression of general neuronal markers such as *Tubb3* (βIII-tubulin) and *Elavl3* (HuC) (Fig. 1b), as well as *Actl6b*, *Map2*, *Gap43* or *Uchl1* (PGP9.5) (Supplementary Fig. 1) (Cl.1–15, 1534 cells; Fig. 1b, Supplementary Data 1, 2). Cl.18 (75 cells) was identified as HCs based on the expression of well-known markers, including *Atoh1*, *Otof*, *Gfi1*, *Pou4f3*, *Cib2*, *Xirp2*, and *Myo6* (Fig. 1b and Supplementary Fig. 1) as well as *Cxcl14*, previously reported in postnatal cochlear HCs (Supplementary Fig. 1)[19,20]. Finally, Cl.19 expressed *Pou3f4* and *Tbx18*, which are known to be enriched in OM cells (Supplementary Data 2)[9,21]. *Tbx1* and *Car3* have also been shown for the OM and are expressed in our dataset[22,23], together with additional gene markers for the OM population, amongst which *Prrx1* and *Twist1* (Fig. 1b, 1145 genes found differentially expressed in OM cluster compared to the E16.5 SGN clusters, Supplementary Fig. 1). This suggests that Cl.19 is likely an OM contaminant population from the FACS purification. This cluster was removed from the dataset for downstream analysis.

Focusing on the neuronal compartment (Fig. 1c, d), alignment of the time points revealed a neuronal differentiation progressing from Cl.1 to Cl.10, 13, 15 and 17 which at P3 represent specified SGN types as identified by the expression of previously published cell-type specific genes for II-, $I_a$-, $I_b$- and $I_c$-SGNs[1] (Fig. 1f). The top differentially expressed genes for each cluster is provided in Fig. 1e. Of note, the $I_b$ and $I_c$ trajectories almost joined at late time-points due to their high similarity in gene expression. However, overlaying them onto FA embedding revealed two paths leading to these two branches, one (the $I_c$) directly from the unspecialized population (see also Fig. 2a) and the other ($I_b$) going through several intermediate states and also giving rise to $I_a$ and II populations (see "Methods" section). Moreover, along the differentiation trajectories, consecutive developmental time-points partially overlapped (Fig. 1d). This is likely explained by the basal-to-apical gradient of neuronal differentiation in the developing cochlea and the fact that whole spiral ganglia (from base to apex) were sampled. Therefore, a continuum of neuronal differentiation states and branching can be observed in a single developmental time point.

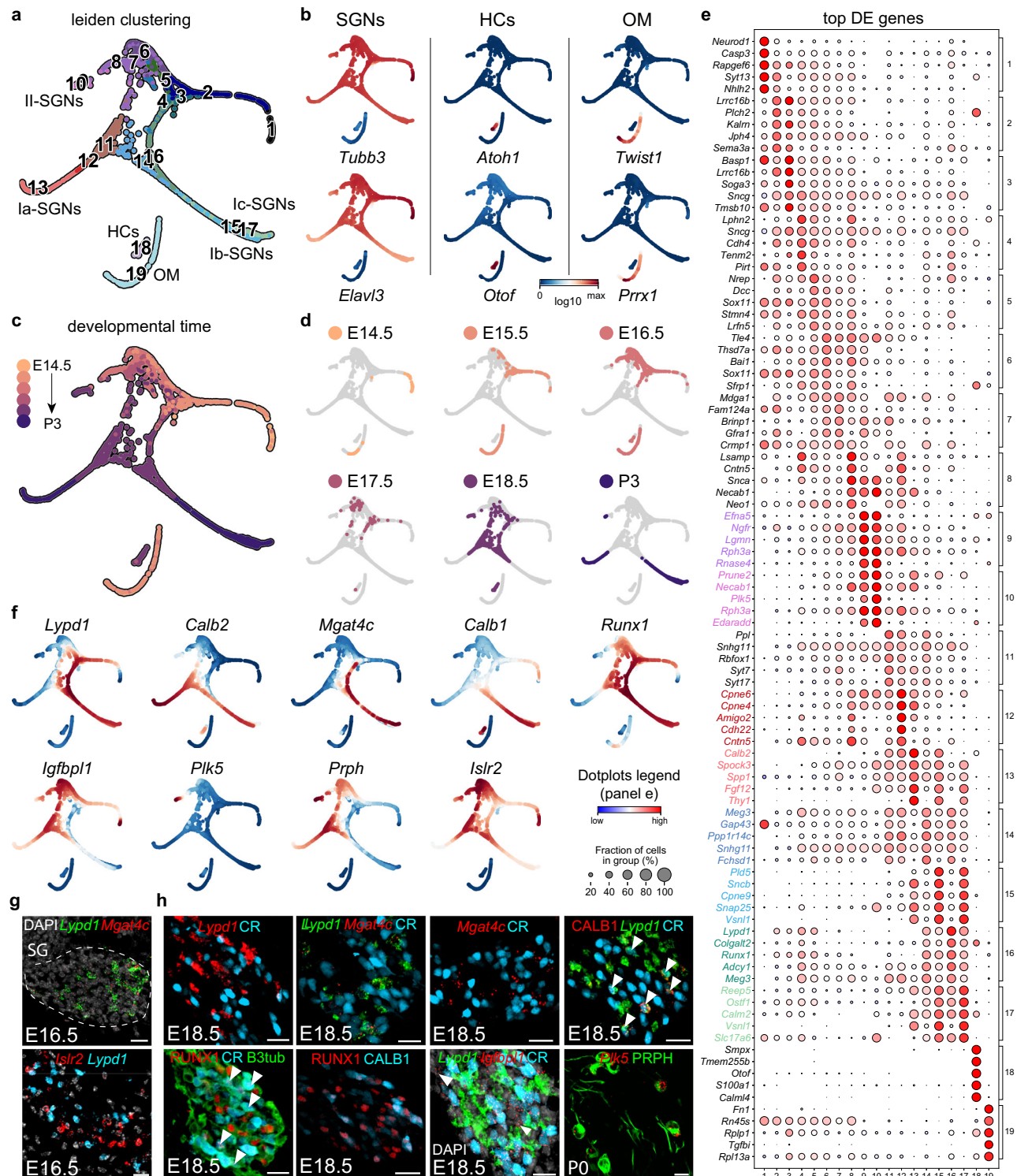

**Fig. 1 Developmental diversification of spiral ganglion neuron lineages. a** UMAP visualization of developing SGNs, showing the trajectory from unspecialized SGNs to distinct subtypes from E14.5 to P3 (Cl.1-17). Cl.18 and 19 denote HC and OM cell clusters. **b** Plots showing expression of marker genes for SGNs, HCs and OM cells. **c**, **d** Plots showing the different developmental time points along the trajectory. **e** Dot-plot of the top 5 most differentially expressed genes (DEGs) for each cluster shown in **a**. **f** Plots showing different marker genes delineating different SGN subtype trajectories. **g**, **h** In vivo confirmation of marker genes from **f** using immunostaining and RNAscope on SG sections from E16.5 (**g**) or E18.5 and P0 (**h**). At E16.5, *Lypd1* and *Mgat4c* colocalize and label the emerging I$_c$ population, while the I$_a$/I$_b$/II lineage is marked by *Islr2* expression. At E18.5, RUNX1, *Lypd1* and *Mgat4c* colocalize and label the I$_c$ population; SGNs expressing only CR (calretinin) represent the I$_a$ population, and SGNs expressing *Lypd1*, CR and CALB1 represent the I$_b$ population. Type II SGNs are marked by *Igfbpl1* and by *Plk5*/PRPH expression at E18.5 and P0. Scale bars: 20 μm. Data in **b** and **f** are MAGIC imputed log10(fpm) expression. DE differentially expressed, HC hair cell, OM otic mesenchyme, SGN spiral ganglion neuron.

The cell state/type specific marker expression was validated in situ using RNAscope and immunostaining (Fig. 1g), confirming at E16.5 the emergence of a $I_c$ identity ($I_c$:*Mgat4c*$^+$ and *Lypd1*$^+$; $I_a$/$I_b$/II: *Islr2*$^+$) and at E18.5 or P0 (Fig. 1h) the existence of the three main subtypes of I-SGNs [$I_a$: CR$^+$ (CR for calretinin); $I_b$: *Lypd1*$^+$/CR$^+$ and *Calb1*$^+$; $I_c$: *Lypd1*$^+$/CR$^-$; at this stage, *Runx1* marks both $I_b$ and $I_c$ and *Igfbpl1*, the $I_a$, $I_b$ and II] and the II-SGNs [*Plk5*$^+$ and PRPH$^+$ (peripherin)]. Overall, this progression recapitulates with the selected experimental time points the dynamic of transcriptional changes occurring in SGNs from E14.5 to P3.

**Molecular codes of neuronal diversification.** To study the flux of molecular changes that occur within the neuronal differentiation continuum, we fitted a principal tree in diffusion space, excluding the HC and OM clusters. The tip of the branch that was enriched with E14.5 cells was selected as a root and pseudotime was subsequently calculated as the distance on the tree from that root. The tree was then represented into a dendrogram recapitulating a branched trajectory based on the transcriptional similarity of pseudotime-ordered cells (Fig. 2a). The resulting tree accurately reflected developmental stages and the branching features of the FA representation (Fig. 2b). Plotting gene expression of known neuronal maturation genes, including *Slc17a7* (VGLUT1), *Nsf*, *Syp* (synaptophysin), *Stxbp1* (Munc18-1), *Snap25*, *Grin1* (NMDAR1) and *Nefh* (neurofilament heavy chain) on the tree supported the developmental progression from an immature, unspecialized state to differentiated neuron subtypes (Supplementary Fig. 2d). Other neuronal maturation genes showed instead a cell-type restricted expression pattern (Supplementary Data 1). Interestingly, some genes such as *Cplx2* (complexin 2) and *Stx1a* (syntaxin 1a), which are members of the SNARE family and associated with the synaptic release of neurotransmitters, exhibited a reversed expression pattern with enrichment in immature neurons, during extension of SGN afferents to their targets[24] (Supplementary Fig. 2d), which suggests a role in the expansion of the plasma membrane during axonal growth[25] towards the sensory epithelium. Therefore, the neuronal differentiation tree could be divided into 2 major states, unspecialized *versus* differentiated neurons (the branching tree leading to the known terminal cell types). Notably, the tree showed unspecialized neurons diverging into $I_c$-SGNs and intermediate $I_a$/$I_b$/II-SGNs around E15–16, which marks the period of afferent innervation of HCs[26]. Knowing the first SGNs are generated at E10 in mice, this indicates that neuronal diversification in the cochlea is a relatively late event that might require cell–cell communication with their peripheral target field, but also, that it is initiated prior to E18.5 in mice, i.e., before functional synapses first emerge and correlated activation of SGNs is observed in the cochlea[27–29].

To investigate the dynamic transcriptional changes along the individual segments of the trajectories (Fig. 2a, c–g, dendrograms) and discover transcription and signalling genes with likely functions in SGN differentiation, we used differential gene expression approaches that characterize pseudotime-ordered molecular trajectories (Fig. 2c–i, Supplementary Data 3–6). Focusing on transcriptional regulators (Fig. 2i, j), we observed that the shared, unspecialized neurons' trajectory was characterized by rapid downregulation of genes commonly associated with early neuronal differentiation processes (*Neurod1*, *Nhlh1* or *Ebf2*), and upregulation of genes that are involved in specification events in cochlea and other systems (*Zfhx3*, *Runx1*, *Mafb*, *Meis2*, *Id1* and *Myt1l*) (Fig. 2i, j). Those genes were later found downregulated, restricted to or over-represented in select neuron type trajectories. *Runx1* and *Meis2* for instance were found further increased in the $I_c$-SGNs as they differentiate from the unspecialized state. In

contrast to *Runx1* however, *Meis2* was not maintained in $I_c$-SGNs, and *Runx1* was later found increased also in $I_b$-SGNs (see below) (Fig. 2i, j). *Pou4f1* (BRN3A) showed a similar trend, i.e., maintained or slightly increased in the $I_c$-SGN trajectory, while dropping in the intermediate $I_a$/$I_b$/II population trajectory, which is in line with a recent study[30]. At the opposite, *Gata3*, which increased in the unspecialized neurons, as previously shown[31], was downregulated in all type I SGNs when each population diverged from their shared trajectory (which happens earlier for $I_c$-SGNs) (Fig. 2i, j). This confirms previous observations[31] and suggests that its master regulator function in specifying a generic identity of SGNs[32] might be incompatible with the final differentiation program of type I subtypes. *Gata3* was however maintained in II-SGNs, together with *Mafb* (which in parallel progressively decreases in I-SGNs) (Fig. 2i, j), confirming earlier studies on their expression[1–3,33,34] and known molecular interactions—with MafB acting downstream of GATA3 to regulate auditory synaptogenesis[35].

Following the first branching point (or split), the intermediate $I_a$/$I_b$/II trajectory was primarily characterized by a general maintenance of specific transcription factors (TFs) expression found before bifurcation and the upregulation of *Id1* and of *Gfra1* (GDNF family receptor alpha 1) (Fig. 2h–j). This suggests that this intermediate state is marked by a progression of neuronal specification events that had already started during the last period of the unspecialized neuron trajectory. In contrast, the further distinction between type II and the transient $I_a$/$I_b$ trajectories was marked in the nascent type II neurons by the absence/decrease of expression of TFs specific to the transient $I_a$/$I_b$ population such as *Runx1* or *Shox2* and the upregulation of *Sox9* and *Tshz3* (Fig. 2i, j). Also, and similar to the $I_c$ population, *Rora* (retinoid-related orphan receptor alpha) and *Prox1*, a generic marker of adult type I SGNs[1], were both upregulated in transient $I_a$/$I_b$ SGNs (Fig. 2i, j). This suggests that *Prox1* could be part of a terminal differentiation program[36] that would be continuously required to maintain a type I specific differentiated state. Finally, the last branching event identified $I_a$- and $I_b$-SGN differentiation coinciding in the $I_b$ trajectory with an upregulation of *Runx1* and *Pou4f1* (although *Pou4f1* showed lower expression relative to $I_c$ neurons), and in $I_a$-SGNs, with a downregulation of *Runx1* and the maintenance, although at lower levels, of *Id1* (Fig. 2i, j). Overall, the data provide a compendium of genes associated with identity divergence, and describe a series of transient, up- and down-regulation of TFs whose expression dynamics may play a significant role in neuronal diversification and cell identity maintenance (see also Fig. 4j).

**Regulon analysis identifies NEUROD1 as a $I_c$ fate regulator in vivo.** Cell type trajectories are shaped by underlying gene regulatory networks (GRNs) that are centred on a limited number of TFs (or master regulators) and co-factors that interact with *cis*-regulatory genomic regions (target genes) to mediate a specialized transcriptional programme that governs individual cell type/state definition. To comprehensively reconstruct GRNs along the neuronal differentiation tree, and reveal the master regulators and co-factors (i.e., regulons) that might govern individual cell type identities, we used SCENIC (Single-Cell rEgulatory Network Inference and Clustering)[37], a computational workflow that enables inference of GRN (regulon) activities. We identified multiple regulons, each representing a TF, along with a set of co-expressed and motif-enriched target genes, and the regulon activity scores for each neuronal trajectory (Fig. 3a). While some of these regulons were shared among multiple trajectories, others were highly specific and non-overlapping and defined the developmental progression of select cell identities. Moreover,

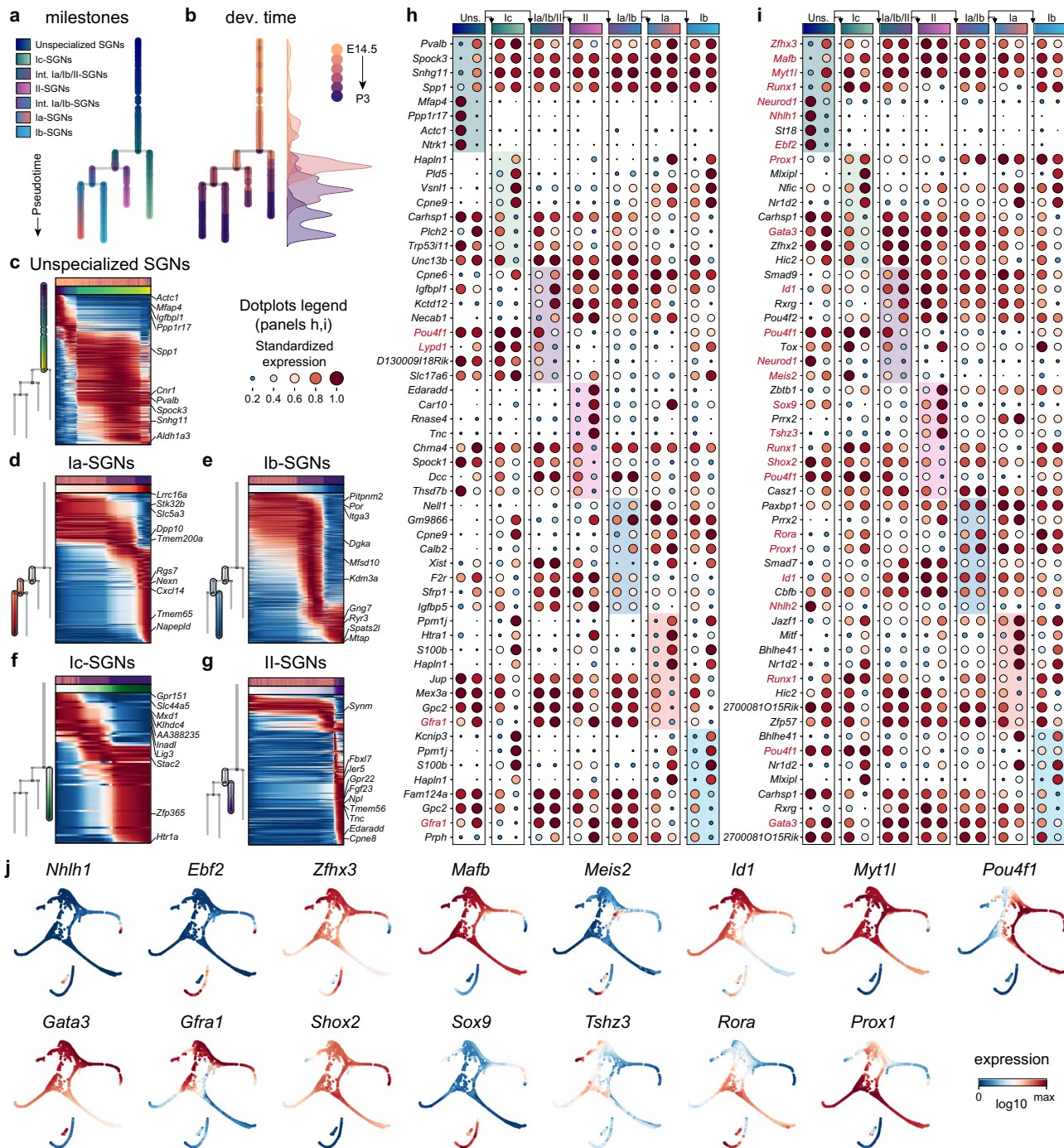

**Fig. 2 Molecular trajectories of developing SGN cell types. a** Dendrogram recapitulating the branched trajectory of the developing SGNs based on the transcriptional similarity of pseudotime-ordered cells. **b** Normalized fraction of cells corresponding to each time point of collection across pseudotime, showing that pseudotime is aligned with age. **c–g** Heatmaps representing overview analysis of the different trajectories, together with the top ten upregulated genes along the differentiation of each branch. **h**, **i** Genes predicted to be likely involved in cell state and cell type divergence along the diversification tree; the top 4 up- and 4 downregulated genes (**h**) and transcription factors (**i**) are shown for each branch/cell state, as defined in **a**. **j** Plots showing different marker genes [in MAGIC imputed log10(fpm) expression] delineating different SGN subtype trajectories. Genes referenced through the text are highlighted in red. The color bars at the top indicate cell states as in **a**. Data in **c–g** are expressed in minimax normalized fitted log10(fpm). Dev. developmental, Int. intermediate.

many regulons were found either up- or downregulated, or transiently expressed (e.g., *Pou2f1*(+), *Onecut2*(+) or *Rora*(+)), along the developmental trajectories, highlighting the need of a decrease in specific GRNs activity for driving proper SGN differentiation. Interestingly, the upregulation of *Ppargc1a*(+) (PGC-1α gene network), that is associated with mitochondrial biogenesis[38], in all emerging type I neurons—with a large energy demand later in life[39]—suggests their possible priming with an increased metabolism already before birth. This co-existence and temporal segregation of various GRNs in each trajectory produce thus combinatorial codes for SGN fate decisions during development.

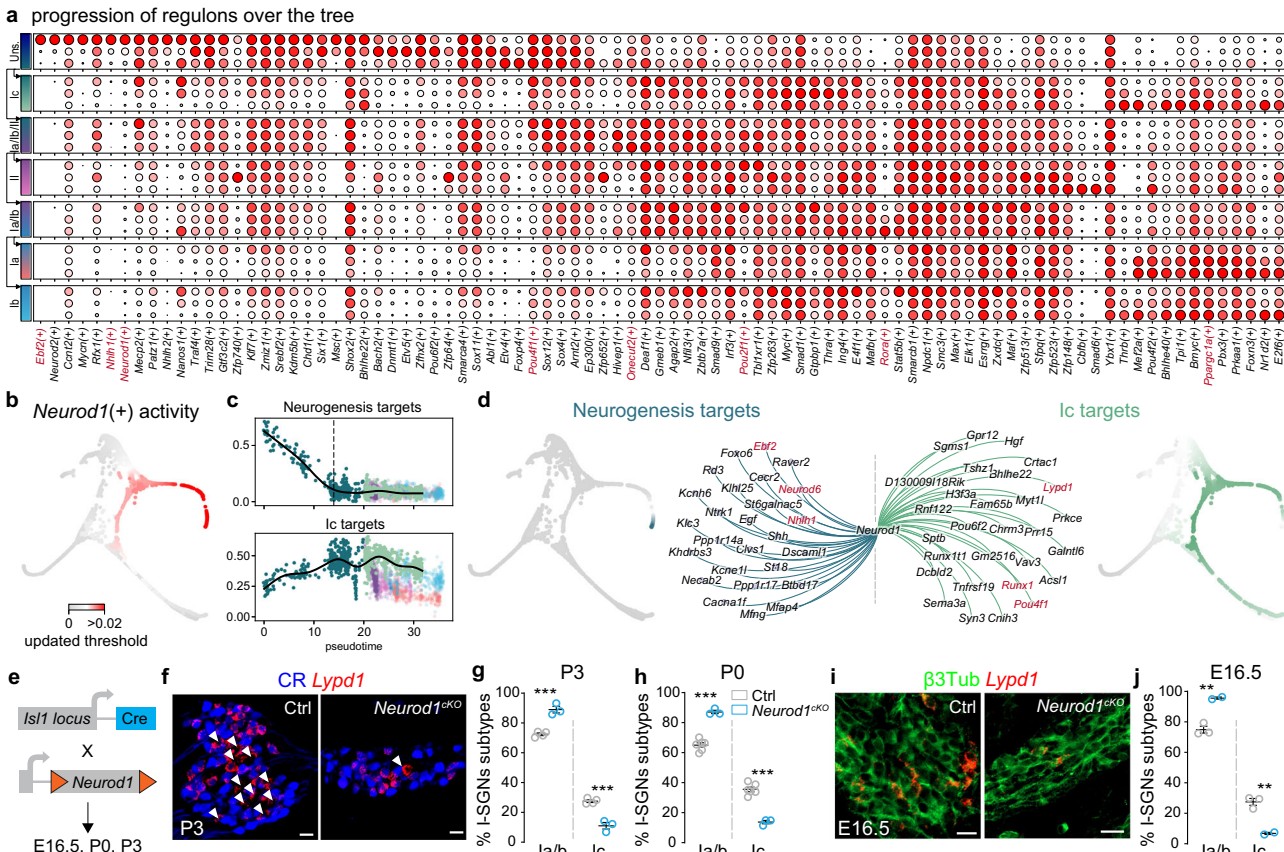

**Fig. 3 Gene regulatory network analysis identifies regulons essential for SGN diversification. a** Dotplot showing the developmental progression of regulons associated with the different cell states and transitions along the diversification tree shown in Fig. 2a; the color bars on the left indicate cell states as in Fig. 2a. **b** Plot showing developmental progression of *Neurod1*(+) after reducing the threshold score. **c** Plots showing developmental (pseudotime) progression of two regimes of activity of *Neurod1*(+) along the differentiation tree. A first regime is active at the beginning of the unspecialized population state, is associated with neurogenesis targets and decreases progressively towards the first bifurcation (representing fate choice between I$_c$ and I$_a$/I$_b$/II); a second regime is progressively active in cells as they reach the first bifurcation and continues to be active in the I$_c$ trajectory. **d** Gene regulatory network representation of *Neurod1*(+) showing its two regimes of activity associated with either neurogenesis (left) or I$_c$ targets (right). **e** Genetic strategy for conditional deletion of *Neurod1* from postmitotic SGNs. **f–j** Neuronal diversification phenotype in cochlea of *Neurod1*^cKO mice at E16.5, P0 and P3. At P3 (**f**, **g**) and P0 (**h**), I$_a$-, I$_b$- and I$_c$-SGNs are CR+, CR+/*Lypd1*+ and *Lypd1*+/CR- (arrowheads in **f**), respectively. At E16.5 (**i**, **j**), SGNs are immunostained for βIII-tubulin and only emerging I$_c$-SGNs express *Lypd1* (see Fig. 1). Quantifications of labelling are shown in **g**, **h** and **j**. Data in **a** represent max normalized fitted AUC scores. Data in **c** represent mean of minmax normalized gene expression from I$_c$ targets and neurogenesis targets, shown as single cell data points as well as gam fit (line) over pseudotime trajectory. Data in **g**, **h** and **j** are presented as mean ± SEM; circles represent values from individual animals (1 cochlea per animal analyzed, minimum of 5 sections per cochlea, basal and mid-basal regions; *n* = 2–6 animals per genotype, per stage); t-test, **\*\*p* < 0.01, \*\*\*p* < 0.001. Source data are provided as a Source Data file. Genes referenced through the text are highlighted in red in **a** and **d**. Scale bars: 20 µm. Ctrl. control.

Our analysis also revealed that the highest representation of regulons defined the unspecialized group of neurons. Because the enrichment and number of effector genes, together with the expression level of the master TF, are key parameters in predicting the activity of a regulon, a potential limitation of the above analysis is the possibility that a regulon might significantly change activity, and thus visibility, depending on the cellular context (co-factors and gene modules involved) such as between two temporally and molecularly separated events, e.g., during neurogenesis (immature state) or neuronal diversification. By re-analysing the activity of neurogenesis-related regulons along the pseudotime axis, we observed that *Neurod1*(+), active in all cells during early specification of the unspecialized SGNs, was also and specifically active at the beginning of the I$_c$ trajectory (Fig. 3b, c). This temporal analysis of *Neurod1*(+) identified two different regimes of *Neurod1*-asssociated GRNs. A first one was associated with the early stage of neuronal differentiation, included targets such as *Nhlh1*, *Neurod6* and *Ebf2*, and decreased along the first,

unspecialized neurons, trajectory (Fig. 3d). A second one progressively increased along this trajectory, diverged to the I$_c$ path, and was marked by the expression of targets including *Runx1*, *Pou4f1* and *Lypd1*, which all characterize the emergence of a I$_c$ identity. To determine if NEUROD1 could influence I$_c$-SGN differentiation, we crossed *Neurod1*^loxP/loxP mice[40] with *Isl1*^Cre mice[41] (ISL1 is expressed from E8.5 in neurons of the developing inner ear[42]) to delete *Neurod1* in SGNs (*Neurod1*^cKO), therefore avoiding the nearly complete loss of SGNs observed in the full *Neurod1* knockout mice[43,44] (Fig. 3e). *Neurod1*^cKO mice have previously been shown to exhibit normal organization of the organ of Corti and to enable development of SGNs at basal and mid-basal regions, albeit in reduced number[45]. Analysis of cochlea in P3 *Neurod1*^cKO mice revealed a loss of I$_c$-SGN marker staining (Fig. 3f, g). This phenotype was similar at P0 and already visible at E16.5 (Fig. 3h–j). To determine if NEUROD1 is required for I$_c$-SGN specification, or if it regulates the survival of this emerging population, or both, we examined cleaved caspase-3

(c-casp-3) immunostaining at E16.5 and P0 in Ctrl and *Neurod1cKO* mice cochlea. Analysis of the basal and mid-basal regions of the cochlea revealed in average about 1 neuron positive for c-casp-3 labelling every two sections in *Neurod1cKO* mice, with no labelling in Ctrl animals (Supplementary Fig. 3). Therefore, while we cannot rule out a potential role of NEUROD1 in early differentiating $I_c$-SGN cell survival, we believe the small presence of SGN cell death observed during the period of neuronal diversification in the absence of NEUROD1 cannot account for the great loss of $I_c$-SGNs seen at E16.5 and around birth. Taken together, these results suggest that NEUROD1 is required for the early specification of $I_c$-SGNs from the unspecialized pool of SGNs.

**Molecular regulation of trajectories and cell states.** Emergence of cell types during development is a continuous process of differentiation where more immature cells progressively become fate restricted in a series of stepwise bifurcation events. These branch points however only indicate an overall change or switch of transcriptional identity along the trajectory, and do not explain the critical events that characterize the pre- and post-transition states and are responsible for the emergence and consolidation of cell fate[46]. To analyse the dynamical behaviour of the hierarchical fate split points that represent diversification (decision) events, we analysed and identified for each branching points gene modules (groups of genes that change in the same direction and tend to synchronize along the pseudotime) possibly driving fate choice (early modules, pre-bifurcation) and fate biasing (late modules, post-bifurcation) before actual fate commitment and during consolidation, respectively. Each bifurcation event was preceded by a period of increasing heterogeneity in cell type-specific module expression, suggesting higher transcriptional differences between the alternative cell fates in each single cell while approaching the bifurcation point which correlated with preference towards a specific fate choice (Fig. 4a–i and Supplementary Data 7). In parallel, the degree of transcriptional coordination within each module increased as cells move toward the bifurcation, together with an increase of the negative correlation inter-modules, thus suggesting co-activation of competing biasing programs prior to fate commitment. Late modules, in contrast, showed mutually exclusive activation in their corresponding branches, consistent with commitment to a particular fate. This analysis was able to identify gene module patterns that were associated to specific cell fate decisions along the diversification tree, extending our previous results on cell state definition. For instance, in the first bifurcation leading to $I_c$-SGNs and the transient $I_a/I_b/II$ population, *Bhlhe22*, *Tle2*, *Rbfox3* and *Pou4f1*, which are known neuronal differentiation TFs, are represented in the $I_c$-early module and therefore likely participate in a fate decision towards a $I_c$ phenotype (Fig. 4a-c and Supplementary Data 7). On the other hand, the $I_c$-late module is marked by the TF *Runx1*, which in other systems, such as the somatosensory neurons, specifies select neuronal identities[47,48], and could here consolidate a $I_c$ fate. In contrast, the early module of the transient $I_a/I_b/II$ trajectory was marked by the expression of many TFs, including the inhibitor of differentiation *Id1*[49] and *Gata3*, which has been described as an intermediate regulator of an auditory neuronal fate[32] (Supplementary Data 7). Interestingly, *Prph* (peripherin), which marks all immature neurons in the cochlea and in the somatosensory system during embryonic development[50,51], was also maintained in the $I_a/I_b/II$ trajectory, confirming the transient, undifferentiated state of this population. Results of this analysis and of previous temporal interrogation of molecular changes and cell state identities are summarized in Fig. 4j.

**Ligand-receptor interactions between HC and SGN types.** To reveal potential communication processes between developing SGNs and their neighbouring cells, we first manually curated genes coding for receptors of well-known pathways that showed changing expression along the pseudotime in our dataset. We noted that, with a few exceptions, most changes in genes associated with morphogens (WNT, BMP, RA and SHH)-, growth factors- and hormones-related signalling were mostly specific to either all type I or type II SGNs (Fig. 5a and Supplementary Figs. 4 and 5). Also, the specific increase of *Nbl1*, *Smurf2* and *Smad6/7/9* in II-SGNs suggests a selective inhibition of BMP signalling in this cell type, as previously suggested[1]. Interestingly, not only receptors and modulators or intracellular signalling components were differentially expressed in SGNs, but also ligands, with for instance *Wnt5a* and *Shh*, which were detected in the type II lineage and the unspecialized population (confirming an earlier study[52]), respectively, suggesting instructive function from the neurons themselves, as previously demonstrated for SGN-derived SHH in HC differentiation[53].

Genes linked to different types of cell–cell adhesion and axon guidance molecules showed, in contrast, a strong cell type trajectory and/or temporal heterogeneity (Fig. 5a and Supplementary Fig. 5). For instance, in the cadherins and protocadherins superfamily, *Cdh23* was found differentially expressed among differentiating type I SGNs at late stages, with higher levels in $I_b$ neurons, *Pcdh17* was initially expressed in immature SGNs and became restricted to the type II trajectory, and *Cdh9* was gradually increased in the transient $I_a/I_b$ and maintained in postnatal $I_a$ and $I_b$ neurons, though with higher levels in $I_a$-SGNs. In axon guidance molecule families, while some genes showed high specificity to specific cell types (e.g., *Epha3* in $I_b/I_c/II$ trajectories or *Epha6* and *Plxna2* in II-SGNs), most genes were broadly expressed but with various levels of expression in distinct trajectories (Fig. 5 and Supplementary Fig. 5). While transcripts at low levels might eventually not be translated or could result in protein levels too low to be functional, these results may also indicate that the construction of the peripheral auditory circuits is at least partly defined by a combinatorial and temporal code of levels of genes, as previously suggested[26]. In this study, the authors showed that, before birth in mice, Semaphorin-3F (SEMA3F, expressed in the OHC region) acts as a repulsive signal for I-SGN processes via the activation of neuropilin-2 (NRP2), and suggested that possible differential expression of NRP2 and/or co-factors (e.g., plexins) might be accounted for the lack of responsiveness of II-SGNs to SEMA3F[26]. In our data, *Nrp2* is progressively upregulated in all I-SGNs from E16.5, but less in II-SGNs (Fig. 5 and Supplementary Fig. 5). Moreover, *Nrp1* levels are at the same time increased in II-SGN trajectory, as well as that of *Plxna2* and *Plxnc1*, suggesting other possible semaphorin signalling, together with select ephrin signalling, acting specifically on II-SGN afferents.

While the distinct innervation patterns of $I_a$-, $I_b$- and $I_c$-SGNs might be temporally regulated, with a gradient of development from the modiolus to the pillar side of the IHCs, a gradient of signalling molecules was also observed, for instance in the axon guidance-related genes, e.g., *Epha3*, *Ntng1/2*, *Plxna4*, *Sema3a/4d*, and *Slit1/2*, but also in the cell adhesion families of molecules, e.g., including cadherins and protocadherins (Fig. 5a and Supplementary Figs. 4 and 5). Overall, our data thus suggest that during their innervation of the sensory epithelium and their central target neurons, the different classes of SGNs might assemble various types of receptors and adhesion molecules in time and space to mediate the selective contributions of diverse attractants and repellents from cells of their surrounding environment (e.g., epithelial cells, HCs or neurons).

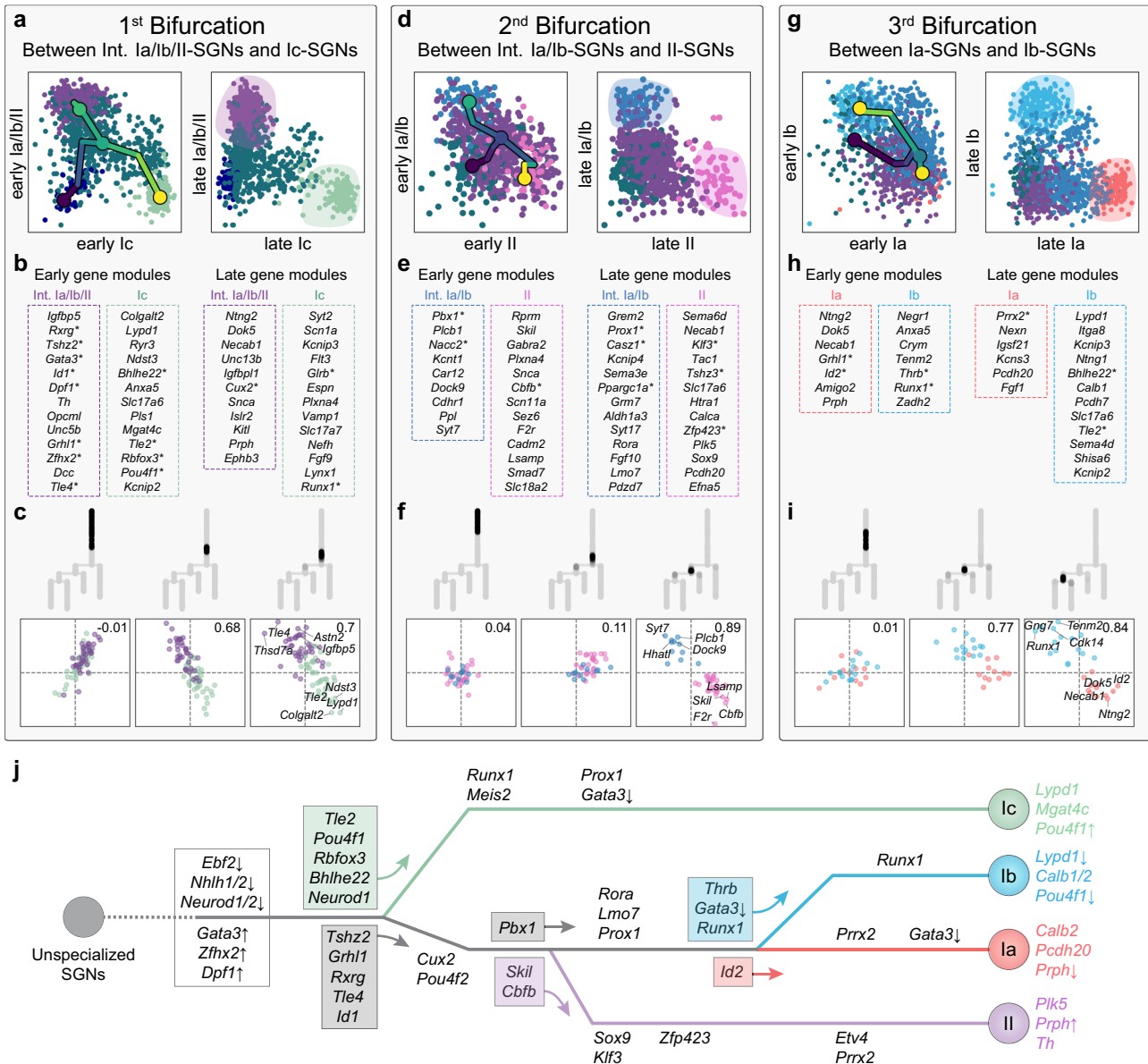

**Fig. 4 Gene modules defining cell fate choice and commitment along the differentiation trajectories. a–i** Analysis of the bifurcations of the differentiation tree representing cell fate selection between distinct neuronal cell lineages: Ic versus Ia/Ib/II (**a–c**), II versus Ia/Ib (**d–f**) and Ia versus Ib (**g–i**). **a, d, g** Scatter plots show average expression of lineage-specific modules in each cell along the trajectory. Early competing modules show gradual co-activation, followed by selective upregulation of one fate-specific module and downregulation of the alternative fate-specific module. Late modules show almost mutually exclusive expression within the two lineages after bifurcation reflecting commitment to either fate. Colors encode tree branches as in Fig. 2a. **b, e, h** Representative genes in each module; asterisks highlight TFs (see also Supplementary Data 7). **c, f, i** Average local correlations of early gene modules with branch-specific correlations, in cells with similar developmental pseudotime (in black on the trajectories); the difference between intra- and inter-module correlations is shown in the upper right corner of the correlation plots and would reflect the repulsion between modules. **j** Summary scheme of the cellular diversification of developing SGNs via distinct trajectories. The transcriptional regulators predicted to be involved in the unfolding of the different lineages are represented on the different trajectories and are derived from the analysis of Figs. 2–4. Genes in colored boxes indicate TFs of early modules from the bifurcation analysis and potentially driving specific cell fate choice (arrows). The arrow symbols after the genes indicate an upregulation or downregulation within a trajectory, while they indicate a high or low expression, respectively, in the final states (Ia, Ib, Ic and II states at the far-right end of the differentiation tree). Int., intermediate.

We next explored the potential receptor-ligand pairing possibilities between the different HC and SGN types during the early stages of HC innervation. To differentiate between IHC and OHC in our dataset, we re-clustered the 39 cochlea HCs of E18.5 from the HC cluster (Cl.18 in Fig. 1a) and obtained two clusters whose gene expression was consistent with the known molecular identity of immature IHCs (*Fgf8*, *Rprm*, and *Trh*) and OHCs (*Bcl11b*, *Scn11a*, and *Insm1*)[12,13] (Fig. 5b–d and

Supplementary Fig. 6). At this early stage of differentiation, a high number of differentially expressed genes (359) distinguished OHCs from IHCs (Fig. 5e and Supplementary Data 8, 9). The relatively low representation of OHCs in our dataset, despite they represent 75% of cochlea HCs in vivo, might be explained by the delayed differentiation of OHCs compared to IHCs[54] coupled to the fact that PV (used to drive Cre expression in our tracing strategy) is a late HC differentiation marker[55]. We then

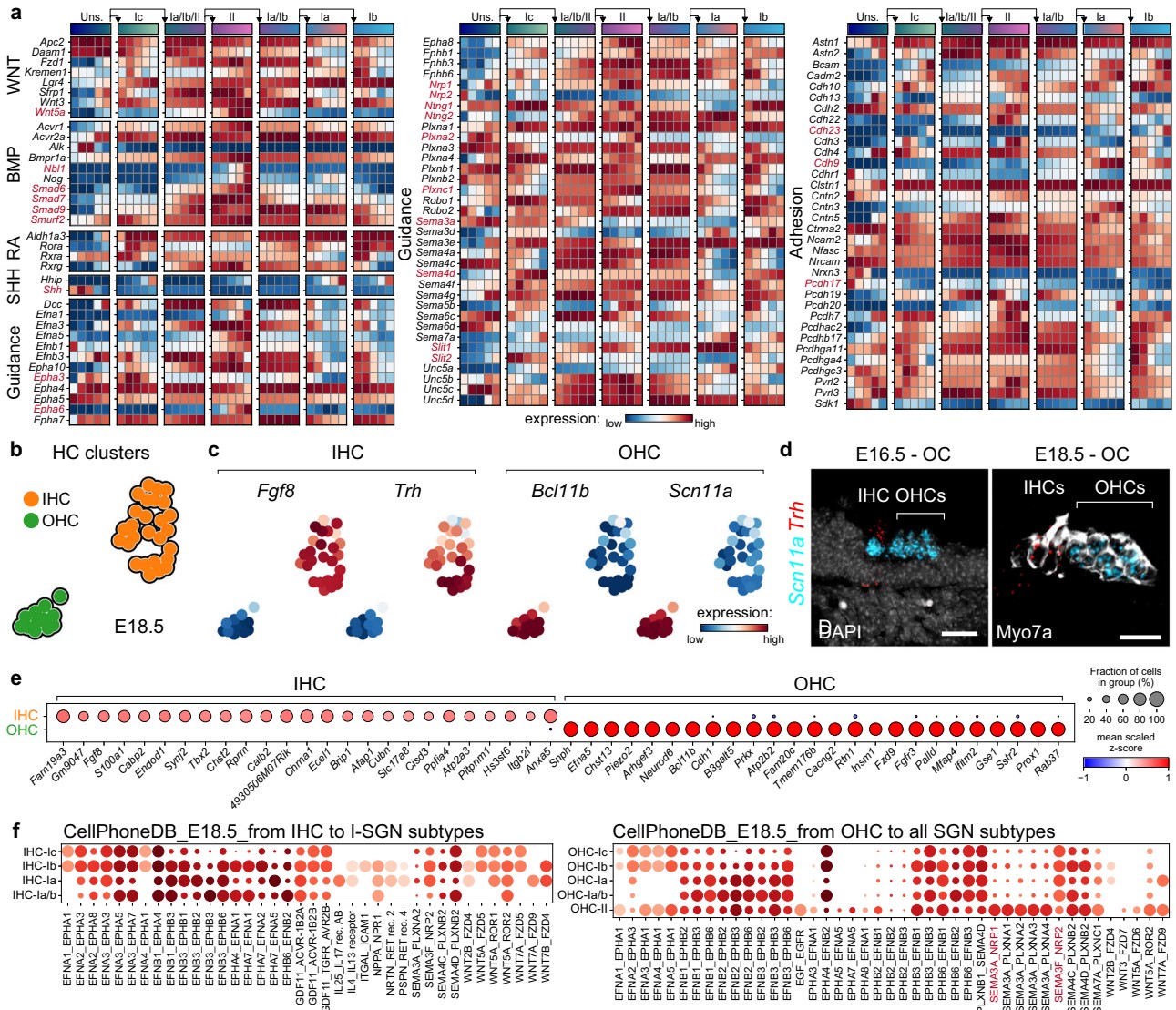

**Fig. 5 Cell–cell communication signatures defining SGN differentiation. a** Expression of genes linked to morphogen signaling, axon guidance and cell adhesion in each neuronal trajectory (see also Supplementary Fig. 4 and 5). The color bars at the top indicate cell states as in Fig. 2a. **b** tSNE plot of the IHC and OHC clusters at E18.5. **c** Representation of selected genes specific to IHC (*Fgf8* and *Trh*) and OHC (*Bcl11b* and *Scn11a*) on the tSNE plot. **d** In vivo confirmation of *Scn11a* and *Trh* expression in HCs at E16.5 and E18.5 using immunohistochemistry and in situ hybridization (RNAscope). Note that *Scn11a* is a general marker for both HC types at E16.5. **e** Dot-plot of the top 25 DEGs between IHC and OHC at E18.5. **f** CellPhoneDB analysis of the top selected potential outgoing cell-to-cell signalling from HCs to SGNs (see also Supplementary Fig. 7). Data in **a**, **c** and **f** represent max normalised fitted gene expression, Knn smoothed log10(fpm) expression and log2 mean of (interacting molecule 1, interacting molecule 2), respectively. Genes referenced through the text are highlighted in red in **a** and **f**. Scale bar: 20 µm. OC organ of Corti.

interrogated potential cell-cell interactions between HCs and SGNs using the permutation-based tool CellPhoneDB[56] that computes a communication score and evaluates the significance of each known ligand-receptor pair (from public resources) found within the scRNAseq data. Importantly, this database considers the multisubunit nature of the protein complexes to identify likely functional ligand-receptor interactions. This method revealed potentially active ligand-receptor pairs from HCs to SGNs and vice versa (Fig. 5f and Supplementary Fig. 7). However, although in other systems instructive signalling from neurons to their peripheral target cells are known to be crucial for the development of the targeted cells[57], both IHC and OHC have been shown to develop in the absence of innervation[58,59], we therefore here only focused on the outgoing HC to SGN signalling as it is more likely to represent an actual signalling network. Moreover, while OHCs are known to interact with I-

SGNs, mostly through repulsive signals[26,60,61], IHC to II-SGN negative interactions are more unlikely to occur at E18.5[60] and were then discarded from the analysis. Also, we filtered ligand-receptor modules based on their expression level and on the number of cells expressing the specific interactors as to provide higher confidence of their biological relevance. Eventually, only cell-type specific ligand-receptor pairs with the highest score are shown (Fig. 5f). More than half of the ligand-receptor pairs involved the ephrin family confirming their important role in shaping cochlear wiring[61–63]. Also, a population of transient $I_a/I_b$ was still observed at E18.5 and showed almost systematically a distinct cell-signalling score with IHCs compared to either $I_a$- and/or $I_b$-SGNs, illustrating a clear parallel between cell type differentiation and their changing pattern of cell–cell communication with HCs. When focusing on OHCs, an interesting observation is the higher score observed for the repellent

SEMA3F-NRP2 signalling from OHCs to I-SGNs, confirming previous observations[26,64], while OHCs to II-SGNs signalling was marked by the SEMA3A-NRP1 pair, which is known to be chemoattractant for neurites[65].

While providing a series of potentially active signalling pathways between HCs and SGNs during a period of important morphological events that characterize the innervation of HCs, these only represent a snapshot of the many possible communications during SGN development and do not consider the possible signalling between SGNs and other cell types. However, because of the depth of our sequencing, these data will be highly valuable to functionally access the molecular basis of the development of the exquisite organization of HC innervation by SGNs.

**Cell-type-specific deafness-associated genes**. To confirm our dataset but also provide insights into the spatio-temporal expression profile of genes associated with hereditary hearing loss (HHL), we extracted HHL-related genes from the Deafness Variant Database (http://deafnessvariationdatabase.org/references) and performed systematic analysis of their expression profile in both developing HCs and SGNs from our databases (Fig. 6a, b and Supplementary Data 10, 11). This analysis identified ninety-two known genes, amongst which about half were either uniquely expressed or enriched in HCs. This indicates that many genes previously associated with HCs function are also potentially expressed, at least transiently, in SGNs. For instance, in our dataset, *Espn* (DFNB36), necessary for late HC stereociliogenesis and postnatal stability/maintenance[66,67], is specifically and progressively enriched in I$_c$-SGNs as they differentiate. Similarly, *Lhfpl5* (also known as TMHS, DFNB66/67), which is critical for HC mechanotransduction[68], is progressively increased in all SGNs during their differentiation, and *Myo6* (DFNA22/ DFNB37), necessary for HC structural integrity[69], is also expressed in all SGNs at early stages.

Our dataset allowed us to also refine knowledge of expression of genes within SGN and HC types. *Pdzd7* (USH2C), *Pnpt1* (DFNB70)[70], *Prps1* (DFNX1)[71], *Tmprss3* (DFNB8/10)[72] and *Wfs1* (DFNA6/14/38)[73], which were known to be expressed in SGNs, showed enrichment in particular subclasses of neurons and/or stages of differentiation. On the other hand, genes known for their expression in other structures of the cochlea were also found in HCs and SGNs. For example, *Tecta* (DFNA8/12/ DFNB21)[74], a major non-collagenous component of the tectorial membrane, was found expressed in IHCs at a surprisingly high level. Moreover, the connexin genes *Gjb2* (Cx26, DFNA3/3A/B1/ 1A) and *Gjb6* (Cx30, DFNA3/3B/1B) which are crucial for the functional differentiation of HCs[75] and yet known to be expressed

in non-sensory cells of the organ of Corti postnatally, were also expressed in IHCs themselves at E18.5 (Fig. 6c), arguing for a potential earlier function of connexins in this cell type.

Together, this analysis uncovers spatio-temporal distributions of deafness genes within developing HCs and SGNs, with some having a transient and cell-type specific expression during embryogenesis. Although deletion or mutations of these genes may not lead to observed dysfunction in either SGNs or HCs in vivo, this updated pattern of expression might help in future studies on cellular mechanisms linked to HHL.

## Discussion

The recent demonstration of diverse molecular types of SGNs has provided a solid basis for the distinct anatomical and physiological properties of auditory afferents[1–3,76], yet how these neuronal identities are generated has remained unexplored. Using scRNAseq-based analysis of developing sensorineural cells of the cochlea, we demonstrate a continuous representation of changes in gene expression that define the course of SGN diversification. We further identify the molecular framework associated with their neuronal fate choices or implementation and cell map expression of genes associated with deafness in developing SGN and HC types. Moreover, our study identifies neuron type specific and temporal differences in expression of chemotropic signaling, notably from HCs, that could potentially act as strong determinants of neuronal differentiation. This work likely contributes very importantly with mechanistic understanding of SGN diversification and will allow researchers to uncover potentially important biological pathways that shape the functional architecture of the primary auditory afferents.

Our study reveals that the fates of SGN subtypes are defined before the first coordinated bursts of action potentials are observed in the auditory nerve, which in rodent occurs perinatally[27–29]. This suggests that SGN diversity does not emerge in response to activity-dependent mechanisms, but instead by specific transcriptional programs that unfold over the course of few days during late embryogenesis. Early spontaneous network activity seen in the postnatal cochlea undoubtedly participates in the maturation and plasticity of the ascending auditory pathways[3,77,78], however most aspects that are directly linked to the functional diversity of SGNs seem to be intrinsically defined before birth. Moreover, unlike the main somatosensory neuron types which differentiate from the neural crest stem cells almost immediately after they exit cell cycle[46], the differentiation of SGNs into subtypes resembles the second phase of neuronal diversification in the somatosensory system in which distinct extrinsic cues act on axons of sensory neurons for generating

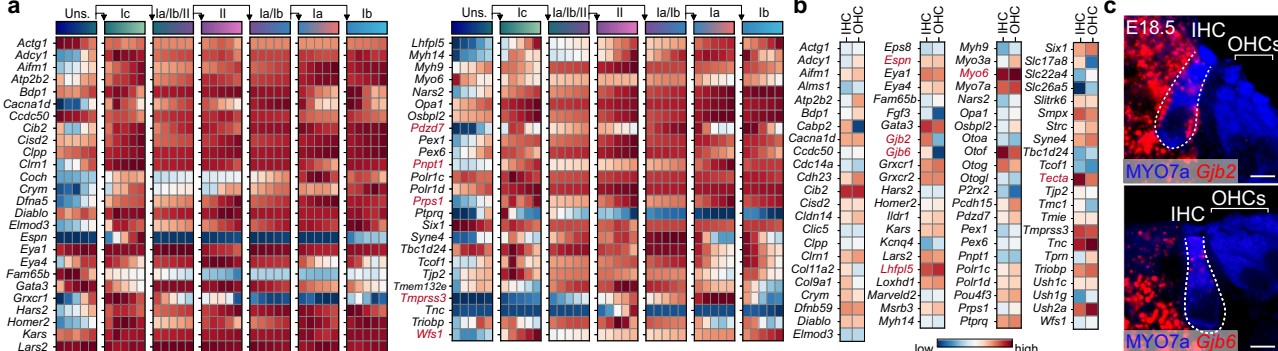

**Fig. 6 Expression of deafness genes in developing SGNs and HCs. a, b** Heatmap of deafness genes related to hereditary hearing loss present in our dataset within SGN lineage trajectory (**a**) as in Fig. 2a and in HC clusters (**b**) from Fig. 5b. Genes referenced through the text are highlighted in red. **c** In situ confirmation of *Gjb2* and *Gjb6* expression in both supporting cells and IHCs (MYO7a+) at E18.5, using RNAscope. Data in **a** represent max normalized fitted gene expression. Scale bar: 10 μm.

further cellular diversity[47,79–82]. It appears that in the inner ear, this transition through subsequent maturation steps, during which neuronal cells acquire different transcription factor networks, is a relatively long process. It is in this context that NEUROD1, which is essential with other co-factors for the development of a spiral ganglion neuron fate during neurogenesis[43,83], was later found transiently expressed and necessary for the emergence of a $I_c$ phenotype through certainly the recruitment of a distinct gene regulatory network. In the developing cochlea, this timing of diversification coincides with the innervation of the sensory epithelium by the unspecialized SGNs[7]. IHCs and OHCs, which can already be transcriptionally distinguished at E14.5 in mice[13], are important candidate cell types for releasing extrinsic signals essential for the differentiation of SGNs into subclasses. While this is well illustrated by the many potentially active cell-cell signaling cassettes between HCs and SGNs in this study, cell-cell communication with OM[9], surrounding glial cells and non-sensory cells of the developing organ of Corti will need to be further studied by cross-comparison computational analysis with recent[13] and future single cell data. Indeed, although a possible spatial (lateral polarity, pillar *versus* modiolar sides), subcellular localization of signaling in the developing IHCs might play a crucial role in shaping neuronal identities and connectivity within the type I neurons, direct or indirect interactions with pillar cells (facing the $I_a$-type) or cells of the Kölliker's organ (facing the $I_c$-type) for instance might also contribute to SGN differentiation.

Another interesting aspect of this study is the developmental history of SGN type lineage. In addition to the comprehensive insights it provides into the molecular programs of SGN differentiation, it also reveals a basic temporal outline of neuronal diversification in the cochlea wherein a $I_c$ identity differentiates first from a common unspecialized pool of SGNs, followed by differentiation of a type II identity. The $I_b$ identity emerges later in the transcriptional hierarchical tree from a common $I_{a/b}$ lineage, which also leads in parallel to the $I_a$ fate. Hence, we propose that diversification into two main types of neurons that differ in their threshold of sensitivity (high- *versus* low-threshold cochlear neurons) and innervation pattern in birds[84] have preceded the emergence of the type II neurons (and presumably the $I_b$ type) that appeared in mammals. Moreover, we suggest that the $I_a$ identity could represent a default path since no specific regulons have been identified to define their differentiation and the profile of expression of specific TFs such as GATA3 (linked to a general program of auditory neuron development)[32] and of peripherin, commonly associated with all immature neurons of the peripheral sensory system during embryogenesis[50,85], were still found in $I_a$-SGNs around birth. Therefore, the $I_a$ trajectory might represent a relatively plastic differentiation path that could have been amenable throughout evolution to the implementation of genetic programs leading to cell type diversification and innovation, including the II- and $I_b$-SGNs.

## Methods

**Ethics and experimental animals**. All animal care and procedures were performed in accordance with the national guidelines published by the Swedish Board of Agriculture and approved by the local ethics committee of Stockholm, Stockholms Norra djurförsöksetiska nämnd. Mice were housed in groups, with standardized pellet food and water ad libitum and under 12 h light–dark cycle conditions. $PV^{Cre};R26^{tdTOM}$ and $Ntrk3^{Cre};R26^{tdTOM}$ mice were crossed from $PV^{Cre}$ (from The Jackson Laboratory, stock No: 017320, C57Bl/6J background) or $Ntrk3^{Cre}$ (from MMRRC, stock No: 000364-UCD, C57Bl/6J background) and $R26^{tdTOM}$ (Ai14, from The Jackson Laboratory, stock No: 007914, C57Bl/6J background) and used to genetically label SGNs and HCs for scRNAseq experiments. $Neurod1^{loxP/loxP}$ (C57Bl/6 background) was published elsewhere[40] and $Isl1^{Cre}$ was obtained from The Jackson Laboratory (stock No: 024242; C57Bl/6J background). Wild-type C57Bl/6J mice were obtained from The Jackson Laboratory (stock #000664) and used for most experiments unless otherwise specified.

**Tissue collection and preparation**. To obtain embryos, time-mating of the respective mouse line was performed. Pregnancy was verified by performing a plug-check on the following day. Plugs were assumed to occur at midnight, wherefore noon of the plug date was designated as embryonic day 0.5 (E0.5) and the date of birth was considered as postnatal day 0 (P0). For embryos, pregnant females were euthanized on E14.5 ($Ntrk3^{Cre};R26^{tdTOM}$), E15.5, E16.5, E17.5 and E18.5 (all from $PV^{Cre};R26^{tdTOM}$) by $CO_2$. After decapitation, the embryonic and neonatal tissues were processed depending on future use.

**RNA in-situ hybridization and immunohistochemistry**. For E16.5 embryos, the whole heads were processed. For E18.5 embryos to P3 pups, the cochleae were surgically dissected from the temporal bone under a stereomicroscope. Briefly, the skull was exposed and split into two halves by cutting along the ventral and dorsal axis from caudal to rostral. The brain and connective tissue were removed, exposing the inner ear, which was then dissected out. Following dissection, the whole head and cochlea were immediately fixed in fresh 4% paraformaldehyde (PFA, Sigma Aldrich) in PBS for 2 h or overnight (O/N) rolling at 4 °C. After fixation, the tissue was washed three times in PBS for 10–15 min each and incubated in sucrose, rolling O/N at 4 °C. Tissues were cryoprotected in 30% sucrose O/N before embedding in OCT. Frozen tissues were kept at -80 °C until sectioning. Blocs were sectioned at 14–16 µm with a Leica cryostat and the slides, kept at −20 °C until further use.

For immunohistochemistry, sections were air dried for 40 min to 1 h at room temperature (RT). Antigen retrieval was applied by immersing the slides in pre-heated 1× target retrieval solution (Dako) for 30 min. The sections were then incubated O/N at 4 °C in blocking solution (0.5% triton, 10% normal donkey serum (Fischer Scientific) and 0.0125% sodium azide), containing the appropriate concentration of primary antibodies in PBS (pH 7.4). Secondary antibodies Alexa-405, -488, -555, -647 (Life Technologies) were applied at 1:500 for 2 h at RT. After three rinses in PBS, samples were mounted and cover-slipped with fluorescent mounting medium (Dako) for imaging. For primary antibodies, the same concentration was used, 1/500. We used rabbit anti-calretinin (Swant), goat anti-PV (Swant), rabbit anti-calbindin (Swant), chicken anti-RFP (Rockland), rabbit anti-RUNX1 (from Thomas Jessel lab), mouse anti-betaIII-tubulin (Promega), rabbit anti-cleaved-caspase3 (Cell Signaling), goat anti-peripherin (Everest Biotech) and DAPI (Invitrogen).

RNA in situ hybridization experiments were performed using RNAscope®. Paired double-Z oligonucleotide probes were designed by the manufacturer against target RNA and are available from Advanced Cell Diagnostics (Newark, CA). The RNAscope® Reagent Kit (Advanced Cell Diagnostics) was used according to the manufacturer's instructions (kit version 2). Frozen fixed tissue sections were prepared according to the manufacturer's recommendations. Each sample was quality controlled for RNA integrity with a probe specific to the housekeeping gene *Ppib*. Negative control background staining was evaluated using a probe specific to the bacterial *DapB* gene. The following probes were used in this study: Mm-*Islr2*-C1, Mm-*Lypd1*-C1/C3, Mm-*Mgat4c*-C1, Mm-*Igfbpl1*-C1, Mm-*Plk5*-C1, Mm-*Calb1*-C2, Mm-*Scn11a*-C1, Mm-*Trh*-C3, Mm-*Gjb2*-C1, Mm-*Gjb6*-C1, Mm-*Prxx1*-C3, Mm-*Twist*-C1, Mm-*Cxcl14*-C1.

**Image acquisition and analysis**. Images were acquired using Zeiss confocal microscope LSM700, LSM800, LSM880 and LSM800 airy equipped with 5×, 10×, 20× and 40× objectives.

**Single cell isolation**. The same dissociation protocol was used for all stages. First, the presence of tdTomato signal was verified under a fluorescence stereomicroscope. From tdTomato positive animals, the cochlea was carefully dissected from the temporal bone as described above. Upon removal of the cochlear capsule, the spiral ganglia (SG) were dissected and collected on ice in Leibovit'z L-15 medium (Life technologies). For E14.5 and E15.5 tissue samples, the cochlea was cut open, and the modiolus region was dissected out. Tissue samples were then digested in a papain-DNAse solution (1.5 ml of papain at 1 mg/ml, 0.5 ml of DNAse at 0.1%) for 20 min, shaking at 700 RPM. After centrifugation of the partially digested tissue at 400 RCF for 10 min, the dissociation mix-solution was removed, and the cell pellet, gently resuspended in Dulbecco's modified Eagle's medium (DMEM) F-12 (Life technologies). Subsequently, cells were mechanically triturated using fire polished Pasteur pipettes coated with 0.2% bovine serum albumin until a homogenized solution was achieved. To remove residual cell aggregates and to generate a single cell suspension, the cell homogenate was then passed through a 70 µm nylon cell strainer (BD Biosciences). Single RFP$^+$ cells were sorted by fluorescence-activated cell sorting (FACS) into individual wells containing lysis buffer in a 384-well plate. The plates were immediately placed on dry ice and stored at −80 °C before being processed for Smart-seq2 protocol. We used 2–4 animals (4–8 cochlea) per experiment, with several rounds of experiments and plates per time-point.

**Single cell RNA-sequencing**. Smart-Seq2 protocol was performed on single isolated cells by Eukaryotic Single Cell Genomics Facility at SciLifeLab, Stockholm (Supplementary Fig. 8). From the $Ntrk3^{Cre};R26^{tdTOM}$ E14.5 samples, we isolated a total of 135 cells, including 82 neurons and 53 OM cells. From the $PV^{cre};R26^{tdTOM}$ mice we isolated a total of 2139 cells: 229 cells at E15.5 (161 neurons and 68 OM

cells), 661 at E16.5 (580 neurons, 73 OM cells and 8 HCs), 72 cells at E17.5 (71 neurons and 1 HC) and 667 cells at E18.5 (611 neurons and 66 HCs). The P3 transcriptional data were obtained from our previous study[1].

**Generation of count matrices, QC and filtering.** The samples were analyzed by first demultiplexing the fastq files using deindexer (https://github.com/ws6/deindexer) using the nextera index adapters and the 384 well layout. Individual fastq files were then mapped to mm10_ERCC genome using the STAR aligner using 2-pass alignment. Reads where filtered for only uniquely mapped and were saved in BAM file format, count matrices were subsequently produced. Estimated count matrices from all plates were combined into one data object, QC metrics were computed using scanpy function[86]. Cells having more than 1000 detected genes and less than 5% of proportion of ERCC reads were kept. A median of 8469 genes were detected per cell in an initial analysis, and 7851 genes after glial code cleanup (see below) (Supplementary Fig. 9).

**Removal of glial contamination.** First, developmental timepoints were analyzed separately. Genes being expressed in less than 3 cells were removed, count matrices were normalized per cell to a target sum of 1000 reads and then log1p was applied. High variable genes detection was performed using pagoda2 approach via scFates package (pp.find_overdispersed, default parameters). PCA was performed on the scaled matrix of over-dispersed genes (scanpy, default parameters). KNN graph (scanpy, pp.neighbors, n_neighbors = 30,n_pcs = 30, metric = "cosine") was generated from the PCA space, and was used as basis for cluster identification using leiden algorithm (scanpy, tl.leiden, default parameters), as well as UMAP embedding (scanpy, tl.umap, default parameters). First inspections of the expression profiles revealed glial contamination, as some clusters were mirrors of other neuronal clusters, while being positive for both neuronal and glial markers. E18.5 timepoint displayed the greatest amount of glial contamination and was used to calculate the glial code. A gene is considered part of the glial code if its expression has a correlation with *Sox10* expression of more than 0.3. This threshold represents the best trade-off between removing the mirrored clusters while keeping most of the possibly informative genes (1371 genes were removed, see Supplementary Data 12 for gene names and their expression level).

**Alignment of the timepoints and main analysis.** The cleaned datasets were combined into one, and the same pipeline was employed as initial analysis, with different parameters for the KNN graph generation (scanpy, pp.neighbors, n_neighbors = 15, n_pcs = 15, metric = "cosine"). The first 15 PCs and developmental time annotation were then used for aligning the data using Harmony python package[17]. E17.5 and E18.5 were merged into one timepoint as there were too few cells from E17.5, which was affecting the results. Augmented affinity matrix was generated (core.augmented_affinity_matrix, n_neighbors=20). Diffusion maps was then generated from this affinity matrix (Palantir, run_diffusion_maps, default parameters) and multiscale space was determined (Palantir, determine_multiscale_space, n_eigs=10) (Supplementary Fig. 8). To generate the Force Atlas embedding, a t-SNE embedding was first generated from the multiscale diffusion space (scanpy, tl.tsne, perplexityt = 100, learning_rate = # of cells/12). Second, from the same multiscale diffusion space, a KNN graph was generated (scanpy, pp.neighbors, n_neighbors=30). Force atlas was then generated from the neighbors graph using the tSNE coordinates as initialization (scanpy, tl.draw_graph, init_pos = "X_tsne",maxiter = 500). Finally, for a selection of leiden clusters, cells being far away from their members on the FA embedding were considered doublets. To detect them, pairwise distances were calculated and standardized, cells hazing a *z* score distance of more than 3 were considered doublets. Differential gene expression was performed using scanpy-python package on this corrected count matrix, using Wilcoxon rank-sum test. We separately analyzed hair cells by re-clustering the HC subset via pagoda2 and generated an UMAP embedding of the cells; vestibular hair cells were discarded from the analysis.

**SCENIC analysis.** SCENIC pipeline was performed using the python package pySCENIC. First, the log-normalized count matrix was used as input, combined with a list of known TFs, to generate regulons based on correlation with putative target genes. Second, using the generated adjacency matrix combined with cis-Traget databases (mm10 500bpUp100Dw and TSS ± 10 kbp), the regulons were refined by pruning targets that do not present an enrichment for a corresponding motif of the TF. Third, cells were scored for each regulon with a measure of recovery of target genes from a given regulon.

**Pseudotime tree inference.** These steps were performed using scFates v0.4.0, a python package built in continuity of the crestree R package[87]. Trajectory inference was performed on SGN cells only.

First, 100 principal graphs composed of 400 nodes were generated with a different random initialization at each run, with SimplePPT approach on the multiscale diffusion space (scFates, Nodes = 400, method = "ppt", ppt_lambda = 1000, ppt_sigma = 0.2). While all trees merged the $I_b$/$I_c$ clusters due to their high similarity, overlaying them onto FA embedding revealed two paths leading to this merged branch, one from immature and one going through other biasing (II and $I_a$). This led us to separately construct two trees to capture both $I_b$ and $I_c$ fates. A first principal graph composed of

600 nodes was fitted with SimplePPT approach on the multiscale diffusion space of a subset containing all clusters except 14 and 15 (scFates, tl.tree, Nodes = 600, method = "ppt", seed = 42, ppt_lambda=50, ppt_sigma = 0.15). A second 100-node principal graph, composed of only Cl. 14 and 15 was then fitted with the same parameters. The two graphs were then manually attached by the tips linking $I_a$ trajectory to $I_b$ trajectory. A root was automatically selected on the resulting merged tree, by selecting the tip with the lowest mean aggregated developmental time value (scFates, tl.root, tips_only = True, min_val=True). From the root was then calculated the pseudotime (scFates, tl.pseudotime, default parameters), from which was generated the dendrogram (scFates, tl.dendrogram, crowdedness = 0.2). Note also that a different Seurat based SNN algorithm was used previously to separate with success the two populations $I_b$ and $I_c$ at P3[1].

**Testing for features associated with the tree.** Feature expression was modeled as a function of pseudotime in a branch-specific manner, using cubic spline regression $exp_i \sim t_i$ for each branch independently. This tree-dependent model is then compared with the unconstrained model $exp_i \sim 1$ using F-test. P-values were then corrected for multiple testing, features were considered significant if FDR < 0.0001.

log10(fpm) count matrix was used to test which genes are significantly changing along the whole tree (scFates, tl.test_association, default parameters), with significant genes being fitted using GAM to obtain smoothed trends (scFates, tl.fit, default parameters). Whole tree was also used in combination with SCENIC derived AUC score, to detect significantly changing regulon activities. To do so, AUC scores were tested (scFates, test_association, A_cut = 0.025) and fitted (scFates, tl.fit, default parameters) via GAM.

**Validating $I_b$ and $I_c$ separate trajectories.** To validate that $I_b$ and $I_c$ biased cells can be separately fitted into two different branches (see Pseudotime tree inference section), both populations from E18 and P3 were fitted together with a single curved trajectory in diffusion space using ElPiGraph[88] (scFates, tl.tree, Nodes=30, epg_mu=200). Significantly changing genes along this trajectory were determined by testing for association separately for $I_b$ and $I_c$ cells, then by taking the union of significant genes (scFates, tl.test_association_covariate, A_cut = 0.5, fdr_cut = 0.01). This list of genes was then used for covariate testing, inspired by a recent preprint[89]. Genes were first tested for amplitude using the following GAM model:

$$g_i \sim s(\text{pseudotime}) + s(\text{pseudotime}) : \text{Covariate} + \text{Covariate} \qquad (1)$$

where $s(.)$ denotes the penalized regression spline function and $s$(pseudotime):Covariate denotes interaction between the smoothed pseudotime and covariate terms. From this interaction term, p-values were extracted and then corrected for multiple testing (scFates, tl.test_covariate, fdr_cut=0.1).

Genes were then tested for trend differences, comparing the model described in (1) to the following reduced one:

$$g_i \sim s(\text{pseudotime}) + \text{Covariate} \qquad (2)$$

Comparison was performed with ANOVA and p-values were corrected for multiple testing (scFates, tl.test_covariate, fdr_cut = 0.1).

**Per trajectory analysis.** For the per trajectory analysis, transitions from parts of the tree to endpoints were subsetted from the whole tree (scFates, tl.subset_tree) and a test for significance was reapplied (scFates, tl.test_association, A_cut = 0.3) to obtain genes changing on that part of the trajectory. Genes were considered specific to a trajectory if they were significantly changing exclusively in one of the analyzed trajectories.

**Bifurcation analysis.** Branch specific genes were first detected via amplitude testing using the following GAM model:

$$g_i \sim s(\text{pseudotime}) + s(\text{pseudotime}) : \text{Branch} + \text{Branch} \qquad (3)$$

From $s$(pseudotime):Branch interaction term, $p$ values were extracted and then corrected for multiple testing (scFates, tl.test_fork, fdr_cut = 0.1). Then, each significant gene was tested for its upregulation along the path from progenitor to terminal state, using the linear model $g_i \sim$ pseudotime. Differentially expressed genes were then assigned between two post-bifurcation branches with fdr < 0.05 and defined differences in expression cutoffs (scFates, tl.branch_specific, cutoffs were specifically set for each bifurcations). Finally, pseudotime of activation was estimated by separating the trajectory into 10 bins, and by calculating the relative expression rate at a specific bin:

$$r(b_t) = \frac{f(b_{t+1}) - f(b_{t-1})}{\max(f) - \min(f)} \qquad (4)$$

where $f$(b) is the mean fitted expression at a specific bin, if the rate was higher than a defined threshold, the gene was considered to activate at the pseudotime value of the related bin.

To analyze molecular mechanisms of cell fate biasing, cell composition was approximated by a sliding window of cells along the pseudotime axis. Cells were manually selected in order to represent the different steps of differentiation. The

local gene-gene correlation reflecting the coordination of genes around a given pseudotime t was defined as a gene-gene Pearson correlation within each window of cells (window sizes were 90 cells for bifurcation A, and 50 cells for the others). The local correlation of a gene g with a module was assessed as a mean local correlation of that gene with the other genes comprising the module. Similarly, intra-module and inter-module correlations were taken to be the mean local gene-gene correlations of all possible gene pairs inside one module, or between the two modules, respectively (scFates, tl.slide_cors, default parameters).

**Cell to cell communication**. First, cell to cell communication was performed using CellphoneDB python package[56] using leiden clustering and corrected log10(fpm) matrix as inputs. As this package uses human databases of interactions, genes were converted to human format using biomaRt database. Pipeline was performed on main E18.5 SGn and HC leiden clusters with default parameters, with means and p values for each pair of interaction between two clusters being the output. For the main figure, manual curation of the results was performed using biological knowledge.

**Statistics and reproducibility**. Statistical analyses were performed using Graph-Pad Prism 8. All validations of sequencing results by in situ hybridization and immunostaining were replicated across at least 5 sections from multiple animals. All micrographs are representative images.

**Reporting summary**. Further information on research design is available in the Nature Research Reporting Summary linked to this article.

## Data availability

Raw sequencing data are available on GEO database under accession code GSE165502 (GSE165502), pagoda2 web application can be explored on a browser via the following link: https://adameykolab.srv.meduniwien.ac.at/SGN/. The data file is also available on the Lallemend laboratory website (lallemend-laboratory) or via the link lfaure/SGN. Moreover, all the analyzed data are available by browsing and analysis via the gene Expression Analysis Resource (umgear). Source data are provided with this paper.

## Code availability

Tree fitting, pseudotime and bifurcation analysis was performed using scFates v0.4.0 python package, available via pypi: scFates. All codes and data for downstream analysis are deposited on the following github repo: sgnfates.

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

## Acknowledgements
We thank the Biomedicum Imaging Core (BIC) Facility supported by the Knut and Alice Wallenberg Foundation and the Eukaryotic Single Cell Genomics (ESCG) facility at SciLife Laboratory. This work was supported by grants from: the Karolinska Institutet Strategic Research program in Neuroscience (StratNeuro), the Swedish Research Council (VR), KID funding, Tysta Skolan foundation and the Swedish Brain Foundation (F.L. and S.H.); the Knut and Alice Wallenbergs Foundation (Wallenberg Academy Fellow), Karolinska Institutet and Ming Wai Lau Foundation (F.L.); the Austrian Science Fund DOC 33-B27 (L.F.); the Czech Science Foundation (20-06927S) and the Czech Academy of Sciences (RVO: 86652036) (G.P.). F.L. is a Wallenberg Academy Fellow in Medicine and a MWLC investigator.

## Author contributions
S.H. and F.L. designed and supervised the study. C.P., P.U. and P.F. collected and processed tissue for analysis, with acquisition of data. I.F. and G.P. generated, processed and provided mutant mouse tissues. C.P, L.F., I.A., S.H. and F.L. analyzed and interpreted data. C.P., L.F. and F.L. drafted the figures. S.H. and F.L. wrote the manuscript, with inputs from all co-authors.

## Funding

## Competing interests
The authors declare no competing interests.
