## [Peer Review File · Nature Communications]

Reviewers' Comments:

Reviewer #1:

Remarks to the Author:

The authors used scRNA-seq of spiral ganglion neurons (SGNs) at different developmental timepoints to determine the changes in the transcriptional profile that drive cell fate determination for different SGN subtypes. The manuscript tries to fill a gap in existing literature and provides novel insights into this process by combining varied analytical approaches of the scRNA-seq data that was generated by the authors. While the manuscript is conceptually interesting and provides a good starting point, in the opinion of the reviewer - more experimental work needs to be done prior to considering this manuscript for publication. This story would be considerably more interesting if any of the numerous hypotheses generated could have been tested experimentally. That would also help provide validation to the data and their bioinformatic approaches. The review below focuses on the conceptual and experimental aspects of the manuscript and not the bioinformatic aspects - as the latter use the common software packages that are standard in the field.

Major concerns:

- 1 - The reviewer notes that the authors generated a relatively small dataset (less than 1600 neurons), of one condition (wild type), that also consists of a total of 40 hair cells (split across the three time points). This is a very small dataset and does not include any replication cohorts or any form of tissue validation.
- 2 - Most importantly, as the authors note, neuronal development begins around E12. It is unclear why the authors chose to have their first developmental time point at E16.5 and not E12 or 14. Emerging data presented in national meetings suggests that those early time points are critical.
- 3 - The manuscript consists of a purely in silico analysis without cross validation to other datasets or generation of replication cohorts. Additionally, the informatic analysis, while using the currently common used tools - do not present a novelty in analysis or application. Experimental work - such as functional validation of the proposed hypotheses in neuronal development, is essential for consideration for publication of a limited dataset as presented, in a medium, and certainly a higher impact journal.
- 4 - The deafness gene analysis of the manuscript consists of a very small number of cells (for hair cells, for example) - and is not novel. Multiple manuscripts analyzing the expression of hair cells at the single cell level, at this and other developmental stages, have been published. The assignment of gene expression to neurons is new, however, not validated at the tissue level or correlated to phenotypes.
- 5 - The authors propose that their manuscript can serve as a resource to the community and in particular comment also on the use of the smart-seq2 which profiles the full length of the transcript and not just the 3' end of the genes. However, the authors have not deposited the data - in its analyzed format, in any of the available tools to allow meaningful browsing of the data (e.g., cellxgene, single cell portal, or the gEAR - which is the main tool of the ear field for sharing of multi-omic data in its analyzed form). This is true also for their previous publication about neuronal gene expression in the ear.
- 6 - the authors claim novelty for Cxcl14 as a hair cell marker and Plk5 as a type 2 marker. As listed below - Cxcl14 is not a novel hair cell marker. And Plk5 was previously reported as a type 2 marker by Goodrich et al and is easily identified as such via browsing-ready resources they (and the Decibel group) made their data available through the gene expression analysis resource (umgear.org).

Specific concerns are listed below per chronological order:

- Line 68-69 'dozen of OHCs that in the mature cochlea modulate the output of the IHCs.' - this sentence is misleading as the OHC modulate the excitation of the IHCs by increasing or decreasing the vibration of the basilar member. But certainly do not directly modulate the output of the IHCs.
- Lines 76-81- very vague. Do not inform the reader of results of substance for which they may wish to continue reading the manuscript.
- Line 86 - "we FAC sorted" - we used flow cytometry to sort.
- Line 86 - Were only tdTom positive cells used for the scRNA-seq analysis? If so, then does this mean that the otic mesenchyme (OM) cells are also tdTom positive in the genetic model used? The OM cells are also the most numerous cells in the cochlea but form a very small population in the

data presented here. The authors must explain this.

-With the data presented throughout the manuscript, it is difficult to understand what is different about cluster 12 and 13. Both seem to represent Ib subtype at P3 and any transcriptional profile differences are not apparent in the data presented here. The authors should highlight the reasons why these cells were clustered separately.

-The image in supplementary figure 1c looks like it was cropped with a freehand tool. Considering DAPI staining was used for nuclei, the only black space in the image should have been the region enclosed by the cochlear duct. This image gives the impression that OM is a small patch of cells between the cochlear duct and SGNs - that's far from the truth. The authors should find a better representative image for this data.

-In figure 2, the colors used in the UMAP and tSNE plots must be explained by a legend.

-Line 106: and new marker genes including Cxcl14, a cytokine receptor previously linked to cellular differentiation Cxcl14 was reported as a hair cell marker in several papers including a recent publication by Stefan Heller.

-Results – flow cytometry images not shown. Authors clearly captured additional cell types. Please share those data.

-Lines 118-125 – this type of description is not helpful for the readers. It is impossible to understand what the authors are referring to without closely observing the figure. The figure is hard to follow without the legend. This is true also for many other parts of the manuscript.

-Lines 170-176 - It's difficult for the reader to visualize the description of how the Gata3(+) fitted regulon activity helped in determining the branches A and B. The authors need to add more annotations to the figure 2 and/or make the figure legend more descriptive.

-In figure 2f, only P0 data for Plk5 expression is shown.

-Line 179 mentions that Plk5 "was found to progressively increase in II-SGNs from E16.5 to P3". Data for other time points and a quantitation of the RNAScope puncta must be included to support this statement.

-For figure 2f, there is no mention in the text for the markers other than Plk5, why they were chosen and how the observations inform the authors' interpretations of the data. -The authors must describe this data in the text and explain why they chose these specific markers for immunofluorescence/ RNAScope validation.

-The white lines changing to colored lines in the pseudotime representation for figure 2 and the multicolor line pseudotime representation in figure 3 must be explained in the figure legends or annotated in the figures.

-The annotations and figure legend for figure 2g just say upregulated and downregulated genes. The text in lines 187-191 suggest it shows novel regulons involved in differentiation of SGNs into four different types. The authors must improve the representation of their data so that the reader can match it to the description in the text.

-The figure legend for figure 2 doesn't mention 2h panel.

-In figure 2h, Maf and Esrrg tSNE plots look exactly the same, but Maf is categorized in figure 2g as Ia/c, and Esrrg is categorized as Ia/b/c. The authors must address this discrepancy.

-For Pou4f1, the data interpretation in lines 194-198 makes sense when looking at the tSNE plot in figure 1d but the tSNE plot in figure 2h looks very different. The authors must fix any mistakes that have happened here.

-Lines 194-200 lead the readers to expect Prrx2 and Pou4f2 data in figure 2h, but that is missing from the figure. The relevant data must be shown since it is important to the authors' interpretations.

-Figure 3d-e shows top 10 genes and transcription factors (TFs) for different early or late modules at bifurcations 1 and 2 of branch A. But lines 228-233 describe somewhere between 16 to 64 genes as being important for a particular early or late module at the bifurcation points. The authors' should include a curated list of these genes categorized into different modules in the supplementary data. The data in the excel sheet for figure 3d-e is not very intuitive to interpret.

-Lines 233-236 - Eya4 has been shown to have a role in hearing, though not specifically in type II SGNs. The authors shouldn't ignore that.

-Lines 241-243 lead the reader to expect data for Six2/4 and Dach1 in supplementary figure 1, but it is missing from the figure. The authors should add that data in the figure.

-Lines 246-248 ignore the fact that figure 3d shows Tfcp2l1 along with Lmo1 and Thrb to be part of the late module TFs for Ia/b.

-Lines 252-254 - The sentence should be rephrased to make it clear that the type Ia within branch A bifurcation 2 is expected to be the default program, not Ib. The sentence sounds confusing as it

is written now. While the absence of a late module TF for Ia at bifurcation 2 is appreciated, how can this hypothesis be explained if the early module TFs at bifurcation 2 are different for Ia and Ib? The authors have not addressed this in the discussion section either at lines 454-457.

- The plots in figure 3f-g must be annotated with gene module names that match the earlier part of the figure to make sense to the reader. There is no mention of the plots with the colored dots are below the tSNE plots with the black dots.
- Lines 263-277 - Have the authors taken into consideration the fact that at a single timepoint the difference between SGN maturation at base vs apex could lead to the variation they see in the data? The authors must discuss this in the text.
- In figure 3h, it would be helpful to have E16.5 to P3 labeled along the horizontal axis to better understand the timeline of the bifurcations in fate determination of the SGN types and subtypes.
- Figure 4a suggests *Plxna3* is not downregulated in type II SGNs and hence cannot be considered to be downregulated in all subclasses as suggested in lines 285-289.
- The selection of genes for tSNE plots in figure 4b doesn't match the description of the figure in lines 289-302. Most of the genes mentioned in the text are missing from figure 4b - *Dcc*, *Plxnb1*, *Plxna3*, *Plxnc1*, *Epha4*, *Lrp11*, *Pcdh19*, *Pcdhgc4*, *Robo2*, and *Bex1*. On the other hand, genes in figure 4b have no mention in the text of the manuscript.
- It would also be helpful if the tSNE plots in figure 4b are placed below tables in 4a to match genes from the respective signaling pathways.
- Figure 4d,e show *Scn11a* as an OHC marker not *Insm1* as mentioned in line 308.
- Transcriptional profile differences between IHCs and OHCs have been described in literature as early as E14.5 and by E18.5, there are clear morphological differences too. So why was it "remarkable" for the authors to find transcriptional profile differences between IHCs and OHCs at E18?
- Lines 332-340 mention CellChat analysis to complement Cellphonedb analysis but the data are not shown. The authors must show these data especially since they think that the IL6 pathway found only through CellChat analysis are an important observation.
- OM cells cluster is not shown in figure 5 though mentioned in lines 367-369.
- In figure 5b-c, the gene names shown on the horizontal axis are too small to read and relate to the descriptions in lines 380-396. The authors should present that data in a way that is more easily understood by the reader.
- In line 395, the authors make a strong claim for connexin proteins being expressed in IHCs at E18. To support this claim, the authors must show antibody staining or RNAScope staining.
- Supplementary figure 2 is not mentioned anywhere in the manuscript. Moreover, while the concept of supplementary fig 2 seems to be similar to Fig 2e, the data shown are very different. The authors must explain this data.
- The authors have not made any mention of a paper published in eLife in 2020 by Li et al. that used a bulk RNA-seq approach to chart a developmental timecourse for SGNs and tested some hypothesis based on it. The authors shouldn't ignore a recently published paper that directly relates to their work.
- Figure 2f looks important, but it is only mentioned in passing (with one or two sentences on PLK5)
- Figure 3. These lineage trajectories are interesting, but it would be nice if the authors would show some staining (IHC or RNA scope) to support their proposed model. Plus, the developmental time points should be superimposed on their graphic (in a way that is consistent with stats-backed data).
- For figures like figure 4 and elsewhere, it would be nice to know the staging in addition to the lineage. At this point, the reader is left to guess. Fig. 4G -- some p-values finally!

Other comments -

- The manuscript must be written better as there is a disconnect between the description of the data in the text and the data that is actually shown in the figures. The introduction and discussion sections also seem to have more grammatical errors than other sections of the manuscript. Please edit also for english.

Reviewer #2:

Remarks to the Author:

The manuscript by Petitpre et al., investigates the change in transcriptional landscape of neuronal

fates in the embryonic and neonatal mouse cochlea using single-cell RNA-seq and bioinformatics. They were able to reconstruct the developmental trajectories of SGN cell fates and identify genes and regulator networks that may play an important role in determining cell states and specification. Comparison between transcriptomes of hair cells and SGNs at different time points also allowed them to identify some novel cell-state-specific cell-cell signaling that may play a role in the differentiation and connectivity profile of SGNs. Although there were several studies in the past two years that examined gene expression profiles and diversity of SGNs in neonatal (including one publication from the author) and adult mice using single-cell RNA-seq, no single-cell RNA-seq based study has been done to analyze transcriptional landscape of embryonic and neonatal SGNs. Thus, far less is known about the molecular mechanisms that govern cell states and specification of SGNs during development. The data in the manuscript is high quality and the analysis is appropriate. The study provides interesting and some novel results that will facilitate further characterization of SGN development, physiology and dysfunction.

I only have a few comments and concerns:

1. Microarray-based transcriptomes of developing SGNs have been published before (Lu et al., 2011, *J Neurosci*, 31: 10903). Bulk RNA-seq of SGNs during development has also been published (Li et al., *eLife* 2020;9:e50491). Those studies need to be cited. Although those datasets and the data presented in this study are not directly comparable, comparison of some important genes in different datasets could enhance the conclusion. This might be important since only a handful of genes were (could be) validated using RNAscope and immunostaining in this study.
2. It might be better to compare transcriptional landscape of SGNs with hair cells and OM. Such comparison could offer important clue about how SGNs differ from other cell types in terms of gene regulatory network and specification.
3. Seven mice per time point were used. I assume that the mice may come from one pregnant dam for one time point. It would be better if embryos from different dams were used for each time point. Please clarify that. Many previous publications presented data with at least two biological repeats.
4. Many genes are only transiently expressed during development and may have no function role in hair cell and SGN function and survival. This is based on many loss-of-function studies. Thus, although some of deafness-related genes are detected in SGNs, deletion or mutations of these genes may not lead to dysfunction of SGNs. Although the analysis is appropriate, its importance may need to be toned down.
5. There are several software packages for analyzing single-cell RNA-seq data. Will usage of more commonly used packages such as monocle or dyno change some conclusions? Dyno could help choose the most appropriate tools for trajectory analysis.
6. Removal of glial contamination needs to be better justified. Will inclusion of this cluster for analysis change some conclusions?

Minor comments:

Page 6, line 111: Please define OM in the text. It was only defined in figure caption.

Page 9, line 193: GRNs. Define that.

Figure caption: Line 488 and Line 509. Scale bar should be in μm , not μM .

Reviewer #3:

Remarks to the Author:

The manuscript by Petitpré et al addresses the developmental trajectory of spiral ganglion neurons into distinct functional subtypes at late embryonic and early postnatal stages using single cell RNA-seq technology. The authors describe potential gene regulatory networks that define discrete branches of the lineage tree resulting in SGN subtype specification. They also use their scRNA-seq dataset to describe potential ligand-receptor pairing between SGNs and their hair cell targets and to profile the expression of known hearing loss genes with some potentially unexpected findings. This work follows up on recent studies by others demonstrating the molecular mechanisms underlying SGN diversity in adult mice and the role that neuronal activity plays in shaping this heterogeneity (Shrestha et al., 2018; Sun et al., 2018). The current study is notable because it

suggests that the initial specification of SGN subtypes occurs prior to their innervation and subsequent dependency on active mechanotransduction currents. The analysis of scRNA-seq datasets in this study is rigorous with findings that should be of potential interest to those in the auditory and developmental neuroscience fields. However, as described below, functional validation of these results is lacking and conclusions are based solely on the computational analysis of dynamic changes in gene expression over time. The following is a list of queries that should be addressed to improve the manuscript.

1. The proposed gene regulatory networks mediating SGN subtype specification are based on computational models of transcriptional dynamics in single cells. However, functional experiments (e.g. ATACseq, lineage tracing, or genetic perturbation) to support key aspects of these models are completely lacking from this study. In the absence of any meaningful validation, the conclusions drawn from this work are considered premature.

2. There are several statements in the text that imply functionality from the analysis of single cell transcriptional dynamics. In the absence of mechanistic experiments, the authors are encouraged to refrain from using definitive statements about the functional significance of these highly correlative studies. The following is a list of examples highlighting this point.

a. On lines 28-31 (abstract) the authors state, "Here we used single cell RNA sequencing to reconstruct the developmental trajectories of SGN cell fates and identified genes and gene regulatory networks that contribute to changes in developmental competence and cell states, and in specification of each major cell types." This sentence is misleading as it implies that gene regulatory networks are indeed contributing to the competence and specification of SGN subtypes, whereas no such functional data is provided to support this conclusion. The authors should use qualifiers such as "may contribute" and "in the possible specification" to avoid overinterpretation of their findings.

b. On line 86 (Results) the authors state, "To analyse the molecular changes that define the diversification of SGN and HC types during development...". Once again, the use of the word 'define' implies a functional relationship between molecular changes and SGN diversification. Since this is a correlative analysis, a more accurate term would be 'associate with'.

c. On lines 196-198 (Results) the authors state, "Interestingly, they showed a complete opposite pattern in the Branch A where Prrx2(+) was up-regulated at the expense of Pou4f1(+), thus depicting complementary, non-overlapping patterns of activity along the diversification tree." The inferred relationship between Prrx2 and Pou4f1 suggests that their expression is inversely correlated. However, in the absence of functional studies one cannot conclude that the upregulation of Prrx2 in Branch A comes at the expense of Pou4f1.

3. The majority of in situ images shown in Fig. 2F, with the exception of Plk5, are not described in the text.

Response to reviewers' comments

We are very glad that the reviewers find our study conceptually interesting, of high quality and that the analysis is appropriate and rigorous. We are also very grateful to the three reviewers for their constructive comments which we think have helped us to consolidate the study; because of this, we also now provide functional validation and have improved the readability of our results. Overall, we have performed additional experiments, analysis and revised the paper according to their remarks. Please find below the detailed answers to each point raised.

REVIEWER #1 (Remarks to the Author):

While the manuscript is conceptually interesting and provides a good starting point, more experimental work needs to be done prior to considering this manuscript for publication.

Major concerns:

1 – The reviewer notes that the authors generated a relatively small dataset (less than 1600 neurons), of one condition (wild type), that also consists of a total of 40 hair cells (split across the three time points). This is a very small dataset and does not include any replication cohorts or any form of tissue validation.

We believe that our new dataset (see below) has largely improved our manuscript, and would like to thank the reviewer for the constructive advice. We also apologize for the lack of clarity in the method section in our previous version. We also provide validation of our findings. Please see detailed response below.

Our initial dataset (from the first version) consisted of about 1500 neurons covering three developmental stages, and the differentiation of 4 SGN types, using the Smartseq2 technology, which is a methodology focusing on very deep sequencing of a relatively low number of cells. This technology is very different from the 10X genomics platform, which is used by many other labs. The 10x methodology is giving much less resolution in term of gene detection per cell and requires sampling several thousands of cells for clustering analysis. Also, the 10X platform does not allow as deep analysis of the molecular changes between clusters and across developmental stages. Given the type of computational analysis performed in our study, Smartseq2 was the ideal methodology to use, and fewer cells were therefore generated (yet relatively large for this technology) and in sufficient number to analyse transcriptional changes across developmental states.

To comply with the reviewer's comment, we have, in our revised version, significantly invested in the generation of new datasets from earlier developmental stages (including both E14.5, E15.5) and from E17.5, as well as increased the number of sequenced cells from already existing developmental time-points. Overall, the new dataset has increased to a total of 2308 cells using the Smart-seq2 platform [in total, 1983 neurons, 75 hair cells and 141 otic mesenchymal cells], which enables us to obtain a high number of sequenced cells (considering our focus on a restricted number of cell types) with yet a much higher resolution of individual transcriptomes as compared to droplet-based methods.

Regarding the hair cell, its cluster was only considered at one time-point (E18.5). Given the high efficiency of the Smartseq2 technology, this number of cells is largely sufficient for clustering and differential gene expression analysis. Of note, trajectory analysis was not performed on HC cluster. In comparison with a recent publication focusing on single cell RNAseq analysis of the developing sensory epithelium using the 10X technology (Kolla et al.,

2020, Nat Commun), and where it was found few specific markers for OHCs at P1 for instance, here, our sequencing and analysis reveal many genes specific to OHCs already from E18.5.

Finally, we apologize for the initial wording in our method section regarding the number of animals or cochlea used per time point. There was indeed no mention of the replicates, while this was done. All experiments had initially been done in duplicate or triplicate and now (in new version), also in quadruplicate depending on the time-point. This is now clarified in the method section. Moreover, *in situ* validation is performed and is shown in figures 1, 4, 6 and Supplementary Fig. 1.

2 – Most importantly, as the authors note, neuronal development begins around E12. It is unclear why the authors chose to have their first developmental time point at E16.5 and not E12 or 14. Emerging data presented in national meetings suggests that those early time points are critical.

We agree with the reviewer, and as stated in our answer to the first comment above, we have invested significantly in adding new time-points, including earlier stages. The new data confirm that the diversification of SGNs, while occurring prior to birth in mice, is a relatively late event, with most molecular changes occurring between E15 and E17 (see Fig. 1). We would like once again to thank the reviewer for the critical inputs to our study, as it has improved, with additional time-points and more molecular insights, our general understanding of the emergence of SGN types during embryogenesis.

3 – The manuscript consists of a purely *in silico* analysis without cross validation to other datasets or generation of replication cohorts. Additionally, the informatic analysis, while using the currently common used tools - do not present a novelty in analysis or application. Experimental work - such as functional validation of the proposed hypotheses in neuronal development, is essential for consideration for publication of a limited dataset as presented, in a medium, and certainly a higher impact journal.

We have now performed functional validation of a TF that we previously identified in our initial manuscript, and confirmed in this new version. Indeed, we show that *NEUROD1* regulon activity is associated with a *Ic* identity, and demonstrate *in vivo*, using conditional mutant mice (*Neurod1* deletion in post-mitotic neurons), that this TF is important for the emergence of a *Ic* identity already from E16.5. This phenotype is confirmed at P0 and at P3 (Figure 3). Moreover, a recent study (unpublished) presented by the laboratory of Lisa V. Goodrich is confirming the role of *RUNX1* in consolidating a *Ic* identity (ARO meeting 2020) and therefore also confirms our analysis.

Moreover, the advantage of our analysis is to provide enough quality and depth in the sequencing data for analysing clear and subtle dynamic of gene expression along and between the trajectories. It provides an advantage to reveal the molecular logic of neuronal differentiation. With the new dataset, such analysis has been improved.

4 – The deafness gene analysis of the manuscript consists of a very small number of cells (for hair cells, for example) - and is not novel. Multiple manuscripts analysing the expression of hair cells at the single cell level, at this and other developmental stages, have been published. The assignment of gene expression to neurons is new, however, not validated at the tissue level or correlated to phenotypes.

As explained in the answer to the first comment, the number of cells for HC clustering and analysis is sufficient using Smartseq2 methodology. This is well exemplified by the numerous marker genes found in Fig. 5 and associated Supplementary data. We understand however the

concern of the reviewer regarding the analysis of the deafness genes in the HC clusters. However, as we noted in our manuscript, some genes previously known for their expression in either the tectorial membrane or supporting cells, were also found expressed in HC clusters. We believe that the high resolution of our methodology enabled these new observations. Previous study on HCs have used the 10X technology, with lower resolution, which certainly explains this difference. Regarding the expression of deafness genes in SGNs, this has now been extended to earlier time points and also has been confirmed in situ for Gjb2 and Gjb6. Please see new Fig. 6.

5 – The authors propose that their manuscript can serve as a resource to the community and in particular comment also on the use of the smart-seq2 which profiles the full length of the transcript and not just the 3' end of the genes. However, the authors have not deposited the data - in its analysed format, in any of the available tools to allow meaningful browsing of the data (e.g., cellxgene, single cell portal, or the gEAR - which is the main tool of the ear field for sharing of multi-omic data in its analysed form). This is true also for their previous publication about neuronal gene expression in the ear.

We agree with the reviewer that such data, in their analysed form, should be accessible to the scientific community. This had been done in 2018 for our previous dataset, which is accessible on the Lallemand laboratory webpage from the date of publication (<https://ki.se/en/neuro/lallemand-laboratory>). Data from the present study will be deposited to the gEAR platform. This has already been organized with Dr. Ronna Hertzano. Moreover, we are planning to provide the whole dataset with imputed expression on Pagoda platform on a link that will be accessible to everyone on the Lallemand laboratory webpage. This platform will allow to navigate through the different developmental trajectories (as shown in Figure 1b, f) using gene expression. This does not require any knowledge in single cell data analysis, enabling anyone to search through the analysed dataset. All associated excel files are also cited in the manuscript and can be uploaded as Supplementary data files. The original dataset will be deposited on the GEO database repository.

6 – The authors claim novelty for Cxcl14 as a hair cell marker and Plk5 as a type 2 marker. As listed below - Cxcl14 is not a novel hair cell marker. And Plk5 was previously reported as a type 2 marker by Goodrich et al and is easily identified as such via browsing-ready resources they (and the Decibel group) made their data available through the gene expression analysis resource (umgear.org).

We agree with the reviewer that Cxcl14 has previously been shown to be expressed in postnatal hair cells in mammals. We now show that it is also an early HC marker, before birth in mice. We apologize for the omission and now refer to the published study in postnatal mammalian cochlea (Sheffer DI et al., 2015). For Plk5, we have not claimed it was a new marker gene, but instead a well-known marker. Indeed, we initially wrote: “PLK5, an established type II-specific marker (Petitpre et al., 2018), ...”.

Specific concerns:

Specific concerns are listed below per chronological order:

-Line 68-69 'dozen of OHCs that in the mature cochlea modulate the output of the IHCs.' – this sentence is misleading as the OHC modulate the excitation of the IHCs by increasing or decreasing the vibration of the basilar member. But certainly do not directly modulate the output of the IHCs.

This has now been corrected: “dozen of OHCs that in the mature cochlea modulate the sensitivity and therefore indirectly the output of the IHCs”

-Lines 76-81- very vague. Do not inform the reader of results of substance for which they may wish to continue reading the manuscript.

This paragraph has now been rephrased.

-Line 86 – “we FAC sorted” – we used flow cytometry to sort.

This is now corrected.

-Line 86 - Were only tdTom positive cells used for the scRNA-seq analysis? If so, then does this mean that the otic mesenchyme (OM) cells are also tdTom positive in the genetic model used? The OM cells are also the most numerous cells in the cochlea but form a very small population in the data presented here. The authors must explain this.

This has now been explained: “This suggests that Cl.19 is likely an OM contaminant population from the FACS purification. This cluster was removed from the dataset for downstream analysis”.

-With the data presented throughout the manuscript, it is difficult to understand what is different about cluster 12 and 13. Both seem to represent Ib subtype at P3 and any transcriptional profile differences are not apparent in the data presented here. The authors should highlight the reasons why these cells were clustered separately.

The initial clustering was based on the differential expression of genes between the Cl.12 and the Cl.13. The new analysis, with an increased number of cells at different stages, and therefore new clusters, does not highlight these new clusters anymore in the clustering plots, but still enable the observation and analysis of molecular changes in Fig. 2-6 where different plots show temporal changes in genes and gene activity along the diversification tree. One likely explanation for the existence of the original Cl.12 and 13 could be that they might have represented a base-to-apex developmental progression of differentiation.

-The image in supplementary figure 1c looks like it was cropped with a freehand tool. Considering DAPI staining was used for nuclei, the only black space in the image should have been the region enclosed by the cochlear duct. This image gives the impression that OM is a small patch of cells between the cochlear duct and SGNs - that’s far from the truth. The authors should find a better representative image for this data.

During development, OM cells are indeed found between SG and afferents along the whole cochlea. The Supplementary Figure 1b (previously 1c) was not cropped. This is similar to many illustrative pictures depicting OM cells such as in Coate et al., Fig. 1d (Neuron, 2012).

Right is a panel from Coate et al. (immunostaining for OM cells in green, SGNs in red and sensory epithelium + glial cells in blue. **On the left** is our panel (E16.5, **Suppl. Fig. 1b**), with two probes for OM cells (Twist1 and Prrx1), with the cellular structure of the cochlea tissue visible using nuclear (DAPI) staining. However, if the reviewer refers to some lack of DAPI

positive cells top left and bottom right, we have observed that during the RNAscope procedure, at this early stage (E16.5), the tissue is fragile and the method is very stringent; as a result, some cells of the OM population can regularly be washed out. We have now added an explanation in the Supplementary Figure 1b legend so that the reader is not misled.

-In figure 2, the colors used in the UMAP and tSNE plots must be explained by a legend.

This is now done in the legend of the new figures.

-Line 106: and new marker genes including Cxcl14, a cytokine receptor previously linked to cellular differentiation Cxcl14 was reported as a hair cell marker in several papers including a recent publication by Stefan Heller.

We agree with the reviewer that Cxcl14 has previously been shown to be expressed in postnatal hair cells in mammals. We now show that it is also an early HC marker, before birth in mice. We apologize for the omission and now refer to the published study in postnatal mammalian cochlea (Sheffer DI et al., 2015). Concerning the ref. to S. Heller, we believe the reviewer refers to Cxcl14 expression in HCs from chicken basilar papilla (Janesick A et al., 2021), which are homologous but not identical to the cochlea HCs in mammals.

-Results – flow cytometry images not shown. Authors clearly captured additional cell types. Please share those data.

Flow cytometry images are now shown in Supplementary Fig. 6. Although we've used a high intensity threshold for selecting tomato fluorescent cells, it is possible that few OM cells have passed this selective parameter. Indeed, the plots are showing a continuum of fluorescence, not two separated clusters of cells along the X axis, which make it difficult to draw a clear separation between highly fluorescent cells and cells with a background fluorescence. Given their high number in the developing cochlea, only a small proportion of OM cells passing this threshold might still represent few dozens of cells in each sample, hence hundreds of OM cells in the whole database.

-Lines 118-125 – this type of description is not helpful for the readers. It is impossible to understand what the authors are referring to without closely observing the figure. The figure is hard to follow without the legend. This is true also for many other parts of the manuscript.

The figure 1 has now been revised, with new data and analysis. We now also correlate lineage trajectory with the developmental (physiological) stage in vivo, and we hope that together, this is providing a clearer representation of the data in figure 1. Also, throughout the rest of the figures, we have also changed our presentation format so that the panels can be followed more easily.

-Lines 170-176 - It's difficult for the reader to visualize the description of how the Gata3(+) fitted regulon activity helped in determining the branches A and B. The authors need to add more annotations to the figure 2 and/or make the figure legend more descriptive.

This has now been totally revised, thanks to the reviewer's comments on including earlier time-points in our analysis. There is no need in the new version to use the Gata3 regulon activity to define the origin of the diversification tree. While Gata3 is shown at many occasions in the new version of the manuscript to act as an important TF of a general SGN identity (prior to their diversification), the new analysis (with data from E14-15) is clearly able to define an unspecialized population of SGNs as the origin of the differentiation tree.

-In figure 2f, only P0 data for Plk5 expression is shown.

While potentially interesting for the type II population, the focus on Plk5 has now been removed in the new version of the manuscript, as it did not help understanding the trajectory.

-Line 179 mentions that Plk5 “was found to progressively increase in II-SGNs from E16.5 to P3”. Data for other time points and a quantitation of the RNAScope puncta must be included to support this statement.

As mentioned above, the focus on Plk5 has now been removed in the new version of the manuscript, as it did not help understanding the trajectory.

-For figure 2f, there is no mention in the text for the markers other than Plk5, why they were chosen and how the observations inform the authors’ interpretations of the data. -The authors must describe this data in the text and explain why they chose these specific markers for immunofluorescence/ RNAScope validation.

We apologize for this and have added those staining in Fig. 1 in the new version, with an explanation in the main text and a description in the figure legend.

-The white lines changing to colored lines in the pseudotime representation for figure 2 and the multicolor line pseudotime representation in figure 3 must be explained in the figure legends or annotated in the figures.

The new version of the manuscript has been largely upgraded, thanks to the reviewer’s comments. As a results, figures have changed quite dramatically, with more depth in the analysis and a better presentation, which we think is now increasing the readability of the data.

-The annotations and figure legend for figure 2g just say upregulated and downregulated genes. The text in lines 187-191 suggest it shows novel regulons involved in differentiation of SGNs into four different types. The authors must improve the representation of their data so that the reader can match it to the description in the text.

We agree with the reviewer, and regulon analysis is now presented in a large dotplot in figure 3, where regulon activity can be followed easily in each trajectory.

-The figure legend for figure 2 doesn’t mention 2h panel.

Figure 2 has now been modified.

-In figure 2h, Maf and Esrrg tSNE plots look exactly the same, but Maf is categorized in figure 2g as Ia/c, and Esrrg is categorized as Ia/b/c. The authors must address this discrepancy.

We apologize for any prior mistake; the figure has now been modified and is clearer to the reader.

-For Pou4f1, the data interpretation in lines 194-198 makes sense when looking at the tSNE plot in figure 1d but the tSNE plot in figure 2h looks very different. The authors must fix any mistakes that have happened here.

We apologize for any prior mistake; the figure has now been modified and is clearer to the reader.

-Lines 194-200 lead the readers to expect Prrx2 and Pou4f2 data in figure 2h, but that is missing from the figure. The relevant data must be shown since it is important to the authors’ interpretations.

All data are now shown in a large plot in figure 3.

-Figure 3d-e shows top 10 genes and transcription factors (TFs) for different early or late modules at bifurcations 1 and 2 of branch A. But lines 228-233 describe somewhere between 16 to 64 genes as being important for a particular early or late module at the bifurcation points. The authors should include a curated list of these genes categorized into different modules in the supplementary data. The data in the excel sheet for figure 3d-e is not very intuitive to interpret.

We apologize if this data file was not incorporated in the earlier version of our study. The new version is including the earlier, unspecialized population of neurons and an important bifurcation leading to Ic and the Ia/Ib/II lineage. This has changed some categorization of genes, which are selected based on their differential expression along the trajectories and on p-values, which obviously depends on the individual cells (which now have been increased in number and cover more time-points) that are considered for comparative analysis.

-Lines 233-236 - Eya4 has been shown to have a role in hearing, though not specifically in type II SGNs. The authors shouldn't ignore that.

We indeed recognize the importance of Eya4 in the very early stage of inner ear formation and should have mentioned it in our earlier version. With new data being added in the new version, we have instead opted for not developing this line of observation.

-Lines 241-243 lead the reader to expect data for Six2/4 and Dach1 in supplementary figure 1, but it is missing from the figure. The authors should add that data in the figure.

See above comment.

-Lines 246-248 ignore the fact that figure 3d shows Tfcp2l1 along with Lmo1 and Thrb to be part of the late module TFs for Ia/b.

We apologize for this. Moreover, by including the unspecialized population of SGNs into the new analysis, the visibility of some specific data has now changed due to the increased number of time-points. Also, to provide high confidence in the potential role of the molecular pathways or TF expression in the differentiation of SGNs, we have applied high threshold for considering gene or gene activity in our analysis. As a results, while the above cited TFs are still found increased in a module to Ia/Ib, they have not been included in the new analysis.

-Lines 252-254 - The sentence should be rephrased to make it clear that the type Ia within branch A bifurcation 2 is expected to be the default program, not Ib. The sentence sounds confusing as it is written now. While the absence of a late module TF for Ia at bifurcation 2 is appreciated, how can this hypothesis be explained if the early module TFs at bifurcation 2 are different for Ia and Ib? The authors have not addressed this in the discussion section either at lines 454-457.

We thank the reviewer for this comment and have now changed the text accordingly (page 21).

-The plots in figure 3f-g must be annotated with gene module names that match the earlier part of the figure to make sense to the reader. There is no mention of the plots with the colored dots are below the tSNE plots with the black dots.

We now have changed the figure and make it clearer.

-Lines 263-277 - Have the authors taken into consideration the fact that at a single timepoint the difference between SGN maturation at base vs apex could lead to the variation they see in the data? The authors must discuss this in the text.

We now have earlier time points to fully cover the entire trajectory of the differentiating SGNs. However, we have discussed the maturation aspect of SGNs at base vs apex in the text,

page 6, second paragraph. Also, with the new data from earlier time points being now added to the analysis

-In figure 3h, it would be helpful to have E16.5 to P3 labeled along the horizontal axis to better understand the timeline of the bifurcations in fate determination of the SGN types and subtypes.

We have now changed our representation of the single cell data and trajectory analysis to follow this bifurcation representation all over the paper, so that the reader can easily refer to any other URD-like plot (similar to Fig. 2a and b) to correlate molecular changes to developmental stage. However, adding specific stages might not really help the reader on this summary scheme because we find it difficult to represent any difference between base and apex without confusing the panel. However, we would be happy to follow any suggestion from the reviewer to answer her/his concern.

-Figure 4a suggests Plxna3 is not downregulated in type II SGNs and hence cannot be considered to be downregulated in all subclasses as suggested in lines 285-289.

We apologize for this mistake, which certainly arose through comparative analysis between different data files. We now have changed completely the presentation of the data, so that it provides a clear, unbiased overview of the expression of the signalling molecules that are changing levels across/along all developmental trajectories. This should be more useful to the reader.

-The selection of genes for tSNE plots in figure 4b doesn't match the description of the figure in lines 289-302. Most of the genes mentioned in the text are missing from figure 4b - Dcc, Plxnb1, Plxna3, Plxnc1, Epha4, Lrp11, Pcdh19, Pcdhgc4, Robo2, and Bex1. On the other hand, genes in figure 4b have no mention in the text of the manuscript.

The new presentation of the data should now resolve this issue.

-It would also be helpful if the tSNE plots in figure 4b are placed below tables in 4a to match genes from the respective signaling pathways.

The new presentation of the data is now responding to this request.

-Figure 4d,e show Scn11a as an OHC marker not Insm1 as mentioned in line 308.

Indeed, it should have been written Figure 4d-f, since Insm1 was shown in fig. 4f. It has now been corrected.

-Transcriptional profile differences between IHCs and OHCs have been described in literature as early as E14.5 and by E18.5, there are clear morphological differences too. So why was it "remarkable" for the authors to find transcriptional profile differences between IHCs and OHCs at E18?

This wording was unnecessary, because of its subjective connotation. We were referring to the large difference between IHC and OHC molecular signatures revealed in our dataset. It was based on a comparison with a recent publication (Kola et al., 2020 Nat Commun), who could hardly find (as the authors said themselves), using the 10X platform, specific genes being expressed in OHCs at P1, while P1 is about 2 days at least after E18.5 (our IHC and OHC clusters). We have now changed the text accordingly.

-Lines 332-340 mention CellChat analysis to complement Cellphonedb analysis but the data are not shown. The authors must show these data especially since they think that the IL6 pathway found only through CellChat analysis are an important observation.

With the new analysis, we have decided to focus on CellPhoneDB and to only show highly significant signalling (as per statistic criteria) with potential relevance in vivo. CellChat analysis has not been performed anymore on the new dataset.

-OM cells cluster is not shown in figure 5 though mentioned in lines 367-369.

OM cell cluster is not mentioned anymore in the analysis of this paper, but only in the Suppl data file 2 for those interested in this population. Data will also be deposited online.

-In figure 5b-c, the gene names shown on the horizontal axis are too small to read and relate to the descriptions in lines 380-396. The authors should present that data in a way that is more easily understood by the reader.

Data presentation is now updated.

-In line 395, the authors make a strong claim for connexin proteins being expressed in IHCs at E18. To support this claim, the authors must show antibody staining or RNAScope staining. Data using RNAScope is added in Figure 6 and confirm expression of these connexins in IHCs (yet at lower levels compared to supporting cells of the IHC compartment) and their complete absence in the OHC compartment at E18.5. Also, the text has been slightly changed to tune down the claim.

-Supplementary figure 2 is not mentioned anywhere in the manuscript. Moreover, while the concept of supplementary fig 2 seems to be similar to Fig 2e, the data shown are very different. The authors must explain this data.

Most supplementary files have now changed, and we hope it is clearer.

-The authors have not made any mention of a paper published in eLife in 2020 by Li et al. that used a bulk RNA-seq approach to chart a developmental timecourse for SGNs and tested some hypothesis based on it. The authors shouldn't ignore a recently published paper that directly relates to their work.

Li et al. have analysed (through bulkseq) the transcriptional profile of all SGNs versus HCs at various postnatal and adult stages, as well as at one embryonic stage (E15.5), with the aim to provide differential marker expression between HCs and neurons to develop two new mouse genetic tools to label SGNs. In response to the reviewer's comment, this work is now cited.

-Figure 2f looks important, but it is only mentioned in passing (with one or two sentences on PLK5)

This has now been changed and moved to Fig. 1 with an explanation in the text and in the figure legend.

-Figure 3. These lineage trajectories are interesting, but it would be nice if the authors would show some staining (IHC or RNA scope) to support their proposed model. Plus, the developmental time points should be superimposed on their graphic (in a way that is consistent with stats-backed data).

The figure has been modified with the new added data and provides a clear presentation of the trajectory tree. Validation of the expression of some genes has been shown in other studies (for instance, see Sherrill HE et al., 2019; Appler et al., 2013; Petitpre et al., 2018; Bok et al., 2013; and fig. 1).

-For figures like figure 4 and elsewhere, it would be nice to know the staging in addition to the lineage. At this point, the reader is left to guess. Fig. 4G -- some p-values finally!

This has been modified now in all figures to follow the trajectories. Note that p-values are part of all single cell data analysis but are part of the original computational analysis. Also please note that throughout the whole manuscript, we have used high statistical significance and high threshold for gene expression and gene activity changes for being considered in our analysis, which provides more confidence on the potential role of the molecular changes in the development and organization of cochlear afferents.

REVIEWER #2 (Remarks to the Author):

The data in the manuscript is high quality and the analysis is appropriate. The study provides interesting and some novel results that will facilitate further characterization of SGN development, physiology and dysfunction. I only have a few comments and concerns:

1. Microarray-based transcriptomes of developing SGNs have been published before (Lu et al., 2011, J Neurosci, 31: 10903). Bulk RNA-seq of SGNs during development has also been published (Li et al., eLife 2020;9:e50491). Those studies need to be cited. Although those datasets and the data presented in this study are not directly comparable, comparison of some important genes in different datasets could enhance the conclusion. This might be important since only a handful of genes were (could be) validated using RNAscope and immunostaining in this study.

In response to the reviewer's comment, these works are now cited.

2. It might be better to compare transcriptional landscape of SGNs with hair cells and OM. Such comparison could offer important clue about how SGNs differ from other cell types in terms of gene regulatory network and specification.

We have now added the gene expression analysis of OM cells in Supplementary Data 2.

3. Seven mice per time point were used. I assume that the mice may come from one pregnant dam for one time point. It would be better if embryos from different dams were used for each time point. Please clarify that. Many previous publications presented data with at least two biological repeats.

We apologize for the initial mistake in the text. All data have been done in duplicate, triplicate or quadruplicate depending on the time-points, with three to four animals used experiment. This has now been clarified in the methods section.

4. Many genes are only transiently expressed during development and may have no function role in hair cell and SGN function and survival. This is based on many loss-of-function studies. Thus, although some of deafness-related genes are detected in SGNs, deletion or mutations of these genes may not lead to dysfunction of SGNs. Although the analysis is appropriate, its importance may need to be toned down.

We agree with the reviewer and have now toned down the text accordingly. Also, we now provide in situ validation for both Gjb2 and Gjb6 expression in IHCs, which is shown in Fig. 6.

5. There are several software packages for analyzing single-cell RNA-seq data. Will usage of more commonly used packages such as monocle or dyno change some conclusions? Dyno could help choose the most appropriate tools for trajectory analysis. Removal of glial contamination needs to be better justified. Will inclusion of this cluster for analysis change some conclusions?

We thank the reviewer for suggesting comparing our trajectory tool (published in Science, Soldatov *et al.*, 2019) to other published ones. Regarding Monocle3, in our case, it is irrelevant for two reasons: (1) the trajectory learning is performed on low dimensional UMAP embedding, which we are not using. Moreover, UMAP is highly sensitive to parameters and can lead to quite distorted embeddings, potentially creating wrong trajectories. (2) the underlying tree learning approach between monocle3 and our tool is the same, namely SimplePPT. The advantage in our case is that we can apply this tree learning method on any dimensionality reduction, which is for our dataset the multiscale diffusion space, comprising 8 dimensions.

Concerning the comment on potential glial cluster, we apologize to the reviewer for not being clear enough initially. No glial cluster were detected in our analysis; therefore, we did not exclude any glial cluster. Instead, as previously shown by other labs (Sun *et al.*, 2018, Cell; Milon *et al.*, 2021, Cell Rep), we obtained some cells containing both neuronal and glial markers, indicating a contamination (possibly doublet). See method section.

Minor comments:

- *Page 6, line 111: Please define OM in the text. It was only defined in figure caption.*
This is now done

- *Page 9, line 193: GRNs. Define that.*
It is now defined.

- *Figure caption: Line 488 and Line 509. Scale bar should be in μm , not μM .*
This is now done.

Reviewer #3 (Remarks to the Author):

The analysis of scRNA-seq datasets in this study is rigorous with findings that should be of potential interest to those in the auditory and developmental neuroscience fields. However, as described below, functional validation of these results is lacking, and conclusions are based solely on the computational analysis of dynamic changes in gene expression over time. The following is a list of queries that should be addressed to improve the manuscript.

1. 1. The proposed gene regulatory networks mediating SGN subtype specification are based on computational models of transcriptional dynamics in single cells. However, functional experiments (e.g. ATACseq, lineage tracing, or genetic perturbation) to support key aspects of these models are completely lacking from this study. In the absence of any meaningful validation, the conclusions drawn from this work are considered premature.

To respond to reviewer's comment, we have performed functional validation of a TF that we previously identified in our initial manuscript, and confirmed in this new version. We show that NEUROD1 regulon activity is associated with a Ic identity, and demonstrate *in vivo*, using conditional mutant mice (Neurod1 deletion in post-mitotic neurons), that this TF is important for the emergence of a Ic identity already from E16.5. Phenotype is confirmed at P0 and at P3. Moreover, another laboratory has recently confirmed the role of Runx1 in consolidating a Ic identity (Lisa V. Goodrich, ARO meeting), which also confirms our analysis.

2. There are several statements in the text that imply functionality from the analysis of single

cell transcriptional dynamics. In the absence of mechanistic experiments, the authors are encouraged to refrain from using definitive statements about the functional significance of these highly correlative studies. The following is a list of examples highlighting this point.

a. On lines 28-31 (abstract) the authors state, “Here we used single cell RNA sequencing to reconstruct the developmental trajectories of SGN cell fates and identified genes and gene regulatory networks that contribute to changes in developmental competence and cell states, and in specification of each major cell types.” This sentence is misleading as it implies that gene regulatory networks are indeed contributing to the competence and specification of SGN subtypes, whereas no such functional data is provided to support this conclusion. The authors should use qualifiers such as “may contribute” and “in the possible specification” to avoid overinterpretation of their findings.

We agree with the reviewer and have now tuned down this statement.

b. On line 86 (Results) the authors state, “To analyse the molecular changes that define the diversification of SGN and HC types during development...” . Once again, the use of the word ‘define’ implies a functional relationship between molecular changes and SGN diversification. Since this is a correlative analysis, a more accurate term would be ‘associate with’.

This is now corrected.

c. On lines 196-198 (Results) the authors state, “Interestingly, they showed a complete opposite pattern in the Branch A where Prrx2(+) was up-regulated at the expense of Pou4f1(+), thus depicting complementary, non-overlapping patterns of activity along the diversification tree.” The inferred relationship between Prrx2 and Pou4f1 suggests that their expression is inversely correlated. However, in the absence of functional studies one cannot conclude that the upregulation of Prrx2 in Branch A comes at the expense of Pou4f1.

This sentence has been changed now.

3. The majority of in situ images shown in Fig. 2F, with the exception of Plk5, are not described in the text.

We apologize for this omission. We have now changed their position to figure 1 and added text both in the result section and in the figure legend.

Reviewers' Comments:

Reviewer #1:

Remarks to the Author:

The authors combine multiple analysis tools (WGCNA, SCENIC, PAGA, ARACNE-AP and VIPER) to analyze the scRNAseq from multiple time points (E14.5, E15.5, E17.5, E18.5, P3). It provides larger scale of datasets to study the cell development, physiology and function comparison to previous version. All reviewer comments are addressed by authors. However, there are a few issues that the authors need to clarify prior to publication. A major important point has to do with FAIR data sharing. In the ear field, data are shared in publications via the gEAR (umgear.org) and it is customary to have short links in figure legends to objects in the gEAR. This allows users to get seamless access to analyzed data and increases the usability of the work. The authors actually say that this will be deposited but this was not done and permalinks were not shared as part of figure legends. A second major concern has to do with the fact that more validation could have been done and better care could have been taken to match text to figures. Below are some detailed comments.

Omics-Specific comments:

1. Authors say data will be deposited into the gEAR platform, however no indication of this is present in the manuscript (should be included in the data availability and code reproducibility section of the manuscript alongside gEAR citation).
2. Line 600-601. What is the read coverage for those genes being filtered? Are they low- or high-expression genes?
3. Line 615-618, what are those glial code genes. Are they enriched for certain pathways?
4. Line 623. How much variation is explained by 15 PCs?
5. Line 628-629, supplementary Fig 6 is the sorting plots, not the multiscale space plots. Please correct it.

Other comments:

1. The colors in Figure 1b should be explained with a legend. The reader can only assume that red denotes high expression while blue denotes lower expression but it is not mentioned anywhere.
2. Lines 170 and 448 - An old paper is referenced here to say that spontaneous activity in the cochlea is observed after birth. Please look at more recent literature which shows spontaneous activity in the cochlea before birth.
3. There is again a disconnect between what is described in the text for Figure 2 and the data in Figure 2. These were the discrepancies noticed by this reviewer:
 - a. Line 179 mentions *Fez1* but this reviewer could not find it in the figure.
 - a. Line 179 mentions *Meis2* as being upregulated during specification (Figure 4j also seems to suggest that) but the data show the opposite - it looks like it is downregulated during specification.
 - b. Line 180 mentions *Runx1* increases in type Ic but it seems to increase in Ic as well as Ib - something that the authors have shown in the summarized diagram in Figure 4j.
 - c. Line 184 mentions *Gata3* is downregulated in all type I SGNs (summary in Figure 4j also suggests this) but significant *Gata3* is seen in Ia type SGNs. This also matches the *Gata3* expression reported by others in that it is not a very specific type II marker.
 - d. Line 189 suggests *Mafb* expression matches *Gata3* expression but that doesn't seem to be the case. It seems to be expressed in all populations except the very early unspecialized SGNs.
 - e. Line 195 suggests *Id1* and *Gfra1* expression is maintained in the intermediate Ia/Ib/II trajectory. This seems to be true for *Gfra1* but not for *Id1* which actually looks upregulated following the first bifurcation.
 - f. Line 205 suggests *Runx1* and *Pou4f1* along with *Lypd1* and *Slc17a6* are upregulated after the last branching event giving Ia and Ib type SGNs. *Runx1* seems to be downregulated in Ia and upregulated in Ib. *Pou4f1* is overall downregulated in both Ia and Ib and actually seems to mark the Ic population. *Lypd1* and *Slc17a6* both seem to mark Ib as well as the Ic population at least as per the data in Figure 2.
4. The authors should make note of the expression timeline for *Isl1* so that the readers can better interpret the results from the experiments where *Isl1Cre* is used for conditional knockout of *Neurod1*.
5. The loss of Ic subtype at E16 shown in Figure 3j quantitatively does not match the image in

Figure 3i.

6. What exactly is CR staining in all the Figures with images for immunostaining or RNAScope? Is it calretinin? The authors should be clear about this at least at the first instance of using this acronym.

7. The authors should be more precise in the description of images as to whether immunostaining is done or an RNAScope probe is used for staining.

8. Lines 257-266 propose the hypothesis that the absence of NEUROD1 leading to small amount of cell death cannot account for the greatly reduced numbers of Ic SGNs seen in the knockout at E16 and P0 based on cleaved caspase 3 staining. Where is this data for cleaved caspase staining?

9. Line 291 - Is "Bhlh22" (labeled "Bhlh222" in Figure 4b) "Bhlhe22" as labeled in Figure 4j?

10. Line 299 mentions peripherin marking Ia/Ib/II trajectory but there is no mention of this gene in Figure 4.

11. Line 311 - This reviewer could find data for Smad6/7/9 and Wnt5a in Figure 5 but not for Shh, Nbl1 and Smurf2.

12. In the rebuttal to point 4 in the previous review, the authors mention that they validated deafness related SGN markers Gjb2 and Gjb6. However, these are expressed and validated for supporting cells and HCs, not SGNs.

Reviewer #2:

Remarks to the Author:

The author did an outstanding job to address reviewers' comments and revise the manuscript. In the revised manuscript, they survey ~2300 cells using smart seq2 technique, which provides much deeper coverage than 10X single cell technique, thus the dataset should be representative. Compared to a recent paper focusing on the neurogenesis of spiral ganglion neurons (Sun et al., 2022; Cell Reports <https://doi.org/10.1016/j.celrep.2022.110542>), this manuscript focused on the diversification/fate determination of spiral ganglion neurons. In addition to profile the transcriptomes of each subtype/transition state, the authors also analyzed the transcriptional driving force behind the transcriptome transition and tested Neurod1 in promoting Ic subtype differentiation. They also tried to dissect cell-cell interactions in spiral ganglion neuron subtype differentiation, which is also interesting. The analysis is appropriate and all figures are in high quality. Overall, the manuscript is significantly improved comparing to the previous version. The manuscript is also beautifully written. The study is important for understanding molecular principles that shape their differentiation.

Following are some minor issues:

1, Generally, the figures should be presented in a better way to show the points the authors tried to make. The gene labels are very small in the figures, thus it's difficult to connect their statements in the text to the figures. They should highlight the genes they mentioned in the text.

2, for Fig 1A, cluster 15 and 17 overlapped with each other, even though they represent two distinctive populations Ic and Ib. They explained the reasons for such overlapping in the method section, but they did not mention/explain in the main text, while can cause confusions for general readers. In addition, in their previous publication, Ic and Ib populations formed clearly separated clusters in P3 samples, thus other clustering methods should be tested to see whether their justification still stand.

3, For figures 2,3 & 5, the author used color codes to indicate different cell types/states, which I found difficult to follow. Addition of cell type/state labels would make it much easier.

4, for figure 5, the heatmap should be clustered on genes to show differential expression among subtypes/states rather than ranking them by the gene name. The same should be done for figure 3.

Reviewer #3:

Remarks to the Author:

The revised manuscript has addressed many of the concerns raised in the previous review. The new functional data on NeuroD1 is a nice addition to the study. There are a few relatively minor issues that should be addressed prior to publication.

Major concerns

1. Despite almost two pages of text (from lines 176-210) in the results section, there are no references to the panels in Fig 2. This makes it very challenging for the reader to critically evaluate the data being described.
2. How is the validity of regulons in Fig.3a determined? What is the positive and negative predictive value of this assay? How well do these regulons fit expectations?
3. In Fig. 5a, Ptchd1 is listed as a gene linked to the Shh signaling pathway. Ptchd1 has a similar sterol sensing domain to the Shh receptor, Ptch1, but is not known to be a component of the Shh signaling pathway (Tora et al., 2017. J. Neurosci 37: 11993-12005).
4. There are many grammatical errors in the text that will need to be corrected by the copy editor.

Response to reviewers' comments

We are very glad that the reviewers recognize the larger scale and deep coverage of our dataset, that we significantly improved our manuscript, and that the study overall will provide high quality data that will be important for understanding the molecular principles that shape auditory neuron differentiation.

Please find below the detailed answers to each remaining minor points raised by the reviewers.

REVIEWER #1

The authors combine multiple analysis tools (WGCNA, SCENIC, PAGA, ARACNE-AP and VIPER) to analyze the scRNAseq from multiple time points (E14.5, E15.5, E17.5, E18.5, P3). It provides larger scale of datasets to study the cell development, physiology and function comparison to previous version. All reviewer comments are addressed by authors. However, there are a few issues that the authors need to clarify prior to publication. A major important point has to do with FAIR data sharing. In the ear field, data are shared in publications via the gEAR (umgear.org) and it is customary to have short links in figure legends to objects in the gEAR. This allows users to get seamless access to analyzed data and increases the usability of the work. The authors actually say that this will be deposited but this was not done and permalinks were not shared as part of figure legends. A second major concern has to do with the fact that more validation could have been done and better care could have been taken to match text to figures. Below are some detailed comments.

Omic-specific comments:

1 – Authors say data will be deposited into the gEAR platform, however no indication of this is present in the manuscript (should be included in the data availability and code reproducibility section of the manuscript alongside gEAR citation).

This is now clearly indicated in the data availability section of the manuscript. As previously described, data from the present study are now deposited to the gEAR platform. This was already organized with Dr. Ronna Hertzano, who has initiated the gEAR platform. Moreover, we are planning to provide the whole dataset with imputed expression on Pagoda platform on a link that will be accessible to everyone on the Lallemand laboratory webpage, following acceptance of the manuscript. This platform will offer the unique possibility to navigate through the different developmental trajectories using gene expression. This does not require any knowledge in single cell data analysis, enabling anyone to search through the analysed dataset. All associated excel files are also cited in the manuscript and can be uploaded as Supplementary data files. The original dataset is deposited on the GEO database repository and will be accessible upon acceptance. This is described in the data availability section, which will be updated upon acceptance.

2 – Line 600-601. What is the read coverage for those genes being filtered? Are they low- or high-expression genes?

We have now provided a spreadsheet (Supplementary Data 12) listing the genes being filtered and their corresponding average number of reads in the dataset. Succinctly, 1371 genes with expression highly correlating with Sox10 were removed, of which 850 were considered highly expressed.

3 – Line 615-618, what are those glial code genes. Are they enriched for certain pathways?

Genes of this glial code are now listed in the Supplementary Data 12. However, due to the high number of genes (1371) composing the glial code, pathway enrichment is not relevant in our case. Indeed, above a thousand genes is a very high number for pathway enrichment, because it will favour enrichment for general processes without specific meaning to the cell type being analyzed. Usually, enrichment analysis works on 50-100 genes.

4 – Line 623. how much variation is explained by 15 PCs?

27% of variation are explained by the 15 first PCs. The explained ratio plot from the first 30 shows a sharp elbow at the beginning and a slowly decreasing tail:

The selection of 15 PCs was arbitrary under the assumption and confirmation that our trajectory, the hair cells and otic mesenchyme cells can all be explained by the first 15 axes of variations.

5 – Line 628-629, supplementary Fig6 is the sorting plots, not the multiscale space plots.

Please correct it.

This is now corrected.

Other comments:

1. The colors in Figure 1b should be explained with a legend. The reader can only assume that red denotes high expression while blue denotes lower expression, but it is not mentioned anywhere.

Colour code legend has now been included in the panel 1b.

2. Lines 170 and 448 - An old paper is referenced here to say that spontaneous activity in the cochlea is observed after birth. Please look at more recent literature which shows spontaneous activity in the cochlea before birth.

Most recent references have now been added.

3. There is again a disconnect between what is described in the text for Figure 2 and the data in Figure 2. These were the discrepancies noticed by this reviewer:

Before answering the comments one by one, please note that for improving the readability of data associated with Figure 2, we have added a new panel “2j” in which the expression of all

the genes that are discussed in the associated text is visualized in a FA plot. Also, these genes are now also highlighted in red in panels 2h and i. We think this will ameliorate the readability of our results.

. Line 179 mentions Fez1 but this reviewer could not find it in the figure.

This is indeed correct, and it has been removed from the text.

a. Line 179 mentions Meis2 as being upregulated during specification (Figure 4j also seems to suggest that) but the data show the opposite - it looks like it is downregulated during specification.

In fact, the expression of Meis2 is indeed upregulated in the trajectory of the unspecialized neurons, as shown initially in Fig. 2i. Only later it is downregulated in most neuron types. To avoid confusion here, we have added some text to clarify this. Also, the FA plots in panel j will help visualizing Meis2 expression along the neuronal trajectories.

b. Line 180 mentions Runx1 increases in type Ic but it seems to increase in Ic as well as Ib - something that the authors have shown in the summarized diagram in Figure 4j.

This is indeed the case, but it was already mentioned when we described the differentiation step between transient Ia/Ib and Ia and Ib, line 205 in the previous version. To avoid any confusion, we now added some text.

c. Line 184 mentions Gata3 is downregulated in all type I SGNs (summary in Figure 4j also suggests this) but significant Gata3 is seen in Ia type SGNs. This also matches the Gata3 expression reported by others in that it is not a very specific type II marker.

GATA3 expression has been reported to decrease during late embryogenesis in I-SGNs and to postnatally become a type II marker (Nishimura et al., 2017). This work was performed using immunohistochemistry. Our data confirms this trend (visible in panel i), with the Ia type being the last type I neurons to downregulate it, leaving Gata3 highly expressed only in type II neurons, as previously demonstrated. This is now better visualized in panel j.

d. Line 189 suggests Mafb expression matches Gata3 expression but that doesn't seem to be the case. It seems to be expressed in all populations except the very early unspecialized SGNs.

The panel 2i is in fact showing a downregulation of Mafb in all type I neurons as they differentiate further. We recognize however that the color code for this gene might be misleading. Indeed, for Mafb, the largest difference in expression levels is observed between the unspecialized neurons and the differentiated neurons, which biased the colour coding to the red channels for all actual (but smaller) differences observed later in the diversification tree. This is inherent to this type of presentation, in which you cannot choose different colour coding along the trajectory. This is why we have added a panel 2j for the genes that are discussed, and in which those differences are more visible. Also, this is also the reason we will provide a readable and searchable pagoda file where all genes can be analysed and visualized the same way as in fig. 2j. Pagoda2 also offers more advanced searching tools for genes with similar trends of expression etc. that will be highly useful for the reader, with no prior necessary knowledge in single cell data analysis.

e. Line 195 suggests Id1 and Gfra1 expression is maintained in the intermediate Ia/Ib/II trajectory. This seems to be true for Gfra1 but not for Id1 which actually looks upregulated following the first bifurcation.

We in fact already wrote that Id1 and Gfra1 are both upregulated in the Ia/Ib/II trajectory in the main text, which is shown in panel 2i and is now also visible in panel 2j.

f. Line 205 suggests Runx1 and Pou4f1 along with Lypd1 and Slc17a6 are upregulated after the last branching event giving Ia and Ib type SGNs. Runx1 seems to be downregulated in Ia and upregulated in Ib. Pou4f1 is overall downregulated in both Ia and Ib and actually seems to mark the Ic population. Lypd1 and Slc17a6 both seem to mark Ib as well as the Ic population at least as per the data in Figure 2.

The text has now been adjusted to better clarify our results, and the new panel 2j is now helping the visualization.

4. The authors should make note of the expression timeline for Isl1 so that the readers can better interpret the results from the experiments where Isl1Cre is used for conditional knockout of Neurod1.

This has now been added in the main text with the appropriate reference.

5. The loss of Ic subtype at E16 shown in in Figure 3j quantitatively does not match the image in Figure 3i.

The images in figure 3 are now representative of the results obtained.

6. What exactly is CR staining in all the Figures with images for immunostaining or RNAScope? Is it calretinin? The authors should be clear about this at least at the first instance of using this acronym.

We apologize for this omission and have now added it both in the main text and in the figure legend.

7. The authors should be more precise in the description of images as to whether immunostaining is done or an RNAScope probe is used for staining.

This was actually done already, as we mentioned that both immunostaining and RNAScope had been used. This was both in the main text and in the figure legend. Note that in mice, transcripts (as genes) are by default italicized, with the first letter in capital, while protein names are (usually) in capital letters, regular format. This is conventional and to our knowledge helps in identifying genes or transcripts from proteins in text or figures. Therefore, in our figures, transcripts detected by RNAScope are all (and the only ones) in italic.

8. Lines 257-266 propose the hypothesis that the absence of NEUROD1 leading to small amount of cell death cannot account for the greatly reduced numbers of Ic SGNs seen in the knockout at E16 and P0 based on cleaved caspase 3 staining. Where is this data for cleaved caspase staining?

A new Supplementary Figure 3 has been added to illustrate our staining.

9. Line 291 - Is “Bhlh22” (labelled “Bhlh222” in Figure 4b) “Bhlhe22” as labelled in Figure 4j?

We apologize for this error and have changed to Bhlhe22 in the text and figure.

10. Line 299 mentions peripherin marking Ia/Ib/II trajectory but there is no mention of this gene in Figure 4.

Actually, Prph (Peripherin, defined as such in the main text) was already mentioned in figure 4b, h and j.

11. Line 311 - This reviewer could find data for Smad6/7/9 and Wnt5a in Figure 5 but not for Shh, Nbl1 and Smurf2.

Shh was in the Figure 5 already, within the Shh signalling-associated genes. However, for Nbl1 and Smurf2, we apologize for this omission and thank the reviewer for noticing it. There has apparently been an error during the last revision of the manuscript and these two genes are now in the Figure 5 and in the new associated Supplementary figure 5.

12. In the rebuttal to point 4 in the previous review, the authors mention that they validated deafness related SGN markers Gjb2 and Gjb6. However, these are expressed and validated for supporting cells and HCs, not SGNs.

We apologize if our sentence in our rebuttal was misleading. It was mentioning that regarding expression of deafness genes, this had now been extended to earlier time points for SGNs and also has been confirmed in situ for Gjb2 and Gjb6 (with new fig.). We apologize for the short cut as indeed the way we phrased it might have implied that we confirmed it for SGNs. However, since the reviewer was interested in confirming expression of Gjb2 and 6 specifically in HCs (as it was not expressed elsewhere), we had (wrongly) anticipated that the sentence was clear and accurate. The figure 6 is indeed showing confirmation of Gjb2 and Gjb6 expression in IHCs at E18.5, as initially requested by the reviewer, and it is described in the main text and the associated figure.

REVIEWER #2

The author did an outstanding job to address reviewers' comments and revise the manuscript. In the revised manuscript, they survey ~2300 cells using smart seq2 technique, which provides much deeper coverage than 10X single cell technique, thus the dataset should be representative. Compared to a recent paper focusing on the neurogenesis of spiral ganglion neurons (Sun et al., 2022; Cell Reports <https://doi.org/10.1016/j.celrep.2022.110542>), this manuscript focused on the diversification/fate determination of spiral ganglion neurons. In addition to profile the transcriptomes of each subtype/transition state, the authors also analyzed the transcriptional driving force behind the transcriptome transition and tested Neurod1 in promoting Ic subtype differentiation. They also tried to dissect cell-cell interactions in spiral ganglion neuron subtype differentiation, which is also interesting. The analysis is appropriate and all figures are in high quality. Overall, the manuscript is significantly improved comparing to the previous version. The manuscript is also beautifully written. The study is important for understanding molecular principles that shape their differentiation.

Following are some minor issues:

1, Generally, the figures should be presented in a better way to show the points the authors tried to make. The gene labels are very small in the figures, thus it's difficult to connect their statements in the text to the figures. They should highlight the genes they mentioned in the text.

We have now increased the police format in most panels, included a new panel 2j where all genes that are mentioned in the main text regarding trajectories (in relation to figure 2) are visualized in FA plots, and we have highlighted (in the plots) the genes that we are mentioning.

2, for Fig 1A, cluster 15 and 17 overlapped with each other, even though they represent two distinctive populations Ic and Ib. They explained the reasons for such overlapping in the method section, but they did not mention/explain in the main text, while can cause confusions

for general readers. In addition, in their previous publication, Ic and Ib populations formed clearly separated clusters in P3 samples, thus other clustering methods should be tested to see whether their justification still stand.

We have now added text in the main text (in addition to the method section) to clarify this. Concerning the usage of other clustering methods, a different Seurat based SNN algorithm was used previously to separate the two populations Ib and Ic, the one we used in the present study (leiden) is different, and both methods led to a similar result.

3, For figures 2,3 & 5, the author used color codes to indicate different cell types/states, which I found difficult to follow. Addition of cell type/state labels would make it much easier. This has now been done.

4, for figure 5, the heatmap should be clustered on genes to show differential expression among subtypes/states rather than ranking them by the gene name. The same should be done for figure 3.

This was initially discussed in the lab: which presentation fits best the potential demand. Some readers who are genes-oriented like to see family of genes that are grouped. Others prefer to focus on trajectory and understand which genes are contributing to or are at least highly represented in a particular trajectory. We initially opted for the first representation of the data. We understand that the second add another angle to it and now provide an additional supplementary figure (Suppl. Fig. 5) illustrating differential expression along the distinct trajectories. Please note that in this case, many genes are shared between trajectories and are therefore represented several times, which is not the case when ranking them by gene name.

Reviewer #3

The revised manuscript has addressed many of the concerns raised in the previous review. The new functional data on NeuroD1 is a nice addition to the study. There are a few relatively minor issues that should be addressed prior to publication.

Major concerns

1. Despite almost two pages of text (from lines 176-210) in the results section, there are no references to the panels in Fig 2. This makes it very challenging for the reader to critically evaluate the data being described.

We apologize for the difficulty to read this section of the manuscript. We have now added more references to panels of Fig. 2 in the text. Moreover, we have now increased the police format in most panels (for higher visibility), included a new panel 2j where all genes that are mentioned in the main text where we are discussing the trajectories (in relation to figure 2) are visualized in FA plots. Moreover, we have highlighted (in the plots) the genes that we are mentioning in the main text. This is also now referenced in the text.

2. How is the validity of regulons in Fig.3a determined? What is the positive and negative predictive value of this assay? How well do these regulons fit expectations?

The regulons shown on Fig. 3a underwent a prefiltering process described in the methods part. These were tested to be significantly changing over the tree using a GAM model, with a threshold of 0.025 AUC score of amplitude of change for being represented in the plot. Regarding the expectations, using this value, we have found relevant biological information: for instance, in the case of Neurod1 regulon which we focused on the rest of the figure. Interestingly, one target gene in this regulon, i.e., Runx1, has been recently reported by the laboratory of Prof. Lisa V. Goodrich to consolidate a Ic identity (ARO meeting 2020),

therefore also supporting our analysis. Another study (Sherrill et al., 2019, J Neurosci), led by Prof. Matthew W. Kelley, recently showed that past E16 in mice, Pou4f1 expression becomes restricted to Ic neurons trajectory during late embryogenesis, confirming our regulon analysis (Fig. 2a), and its connection to Neurod1 regulon in Figure 2d. Moreover, as written in the main text, a large number of regulons are represented in the first, unspecialized neuron, trajectory, and most of them are very well known for their role or expression during neurogenesis in other regions of the nervous system, but also in the cochlea such as Neurod1 (Liu et al., 2000, Genes Dev), Klf7 (Laub et al., 2001, Dev Biol; Lei et al., 2001, Development), Pou6f2 (Patthey et al., 2016, Neural Dev), Nhlh1 and Nhlh2 (Jahan et al., 2010, Plos One), or Mycn (Domínguez-Frutos et al., 2011, J Neurosci). Other genes, such as, for instance, Shox2 (Li et al., eLife, 2021), Pou4f1 (Sherrill et al., 2019, J Neurosci), Sox4 and Sox11 (Gnedeva and Hudspeth, 2015, PNAS), Mafb (Yu et al., 2013, eLife), Esrrg (Li et al., 2021, eLife), Pou4f2 (Deng et al., 2014, Gene Expr Pat), have been shown, in contrast, to be expressed in differentiating SGNs, with Mafb being crucial for instance for post-synaptic synaptogenesis of SGNs (Yu et al., 2014, eLife). While all references cannot be cited, many of them are discussed and cited in the main text. We hope the reviewer agrees that our regulon analysis fits well with expectations, are supported by functional studies (including Neurod1 function, present study), and also provides additional knowledge on new potentially important transcriptional regulators of SGN subtypes differentiation.

3. In Fig. 5a, Ptchd1 is listed as a gene linked to the Shh signaling pathway. Ptchd1 has a similar sterol sensing domain to the Shh receptor, Ptch1, but is not known to be a component of the Shh signaling pathway (Tora et al., 2017. J. Neurosci 37: 11993-12005).

We thank the reviewer for noting this error, which has now been correct (by removing the gene from the list).

4. There are many grammatical errors in the text that will need to be corrected by the copy editor.

These have now been corrected throughout the manuscript.

Reviewers' Comments:

Reviewer #2:

Remarks to the Author:

The authors have adequately addressed questions/issues raised by Reviewer 1 and Reviewer 2.

They also revised the manuscript based on comments by these two reviewers. Overall, the manuscript is

significantly improved. The analysis is appropriate and all figures are in high quality. The study is important for understanding molecular principles that shape cochlear spiral ganglion neuron differentiation.

Reviewer #3:

Remarks to the Author:

The authors have satisfactorily addressed all of my previous concerns.